# Beyond Accuracy: Dissecting Mathematical Reasoning for LLMs Under Reinforcement Learning

**Jiayu Wang**[†][*]**, Yifei Ming**[‡][*]**, Zixuan Ke**[‡]**, Caiming Xiong**[‡]**, Shafiq Joty**[‡]**,**
**Aws Albarghouthi**[†]**, Frederic Sala**[†]
[†]University of Wisconsin-Madison      [‡]Salesforce AI Research

## Abstract

Reinforcement learning (RL) has become the dominant paradigm for improving the performance of language models on complex reasoning tasks. Despite the substantial empirical gains demonstrated by RL-based training methods like GRPO, a granular understanding of *why* and *how* RL enhances performance is still lacking. To bridge this gap, we introduce 💥 **SPARKLE**, a fine-grained analytic framework to dissect the effects of RL across three key dimensions: (1) *plan following and execution*, (2) *knowledge integration*, and (3) *chain of subproblems*. Using this framework, we gain insights beyond mere accuracy. For instance, providing models with explicit human-crafted, step-by-step plans can surprisingly *degrade* performance on the most challenging benchmarks, yet RL-tuned models exhibit greater robustness, experiencing markedly smaller performance drops than base or SFT models. This suggests that RL may not primarily enhance the execution of external plans but rather **empower models to formulate and follow internal strategies better suited to their reasoning processes**. Conversely, we observe that *RL enhances models' ability to integrate provided knowledge into their reasoning process*, yielding consistent gains across diverse tasks. Finally, we study whether difficult problems—those yielding no RL signals and mixed-quality reasoning traces—can still be effectively used for training. We introduce 💥 **SparkleRL-PSS**, a multi-stage RL pipeline that reuses hard problems with *partial step scaffolding*, guiding exploration effectively without additional data generation. Together, our findings provide a principled foundation for understanding how RL shapes model behavior, offering practical insights for building more adaptive, data-efficient, and interpretable RL pipelines for reasoning tasks. Our code, data, and checkpoints are available at: `https://sparkle-reasoning.github.io/`.

## 1 Introduction

Reasoning models are among the most exciting recent developments in the large language model space. These models are able to perform mathematical and other forms of reasoning and achieve excellent performance on a number of benchmarks [2, 15, 37, 47]. Reinforcement learning-based training appears to be crucial to achieving these reasoning capabilities, leading to a proliferation of papers proposing new RL-based training techniques, reasoning models, and evaluation benchmarks.

Despite this explosion of interest, precisely what capabilities are gained during RL training is not clear. Most works studying RL for reasoning use a set of standardized benchmarks. While convenient, tracking the gain in accuracy for a method on, for example, the AIME 2024 contest [32], ultimately provides limited signal into what behaviors are enabled by RL. To make further progress, we argue that it is necessary to develop a ***fine-grained understanding*** of the benefits of RL.

---

[*]Co-lead. Correspondence to: milawang@cs.wisc.edu

39th Conference on Neural Information Processing Systems (NeurIPS 2025).

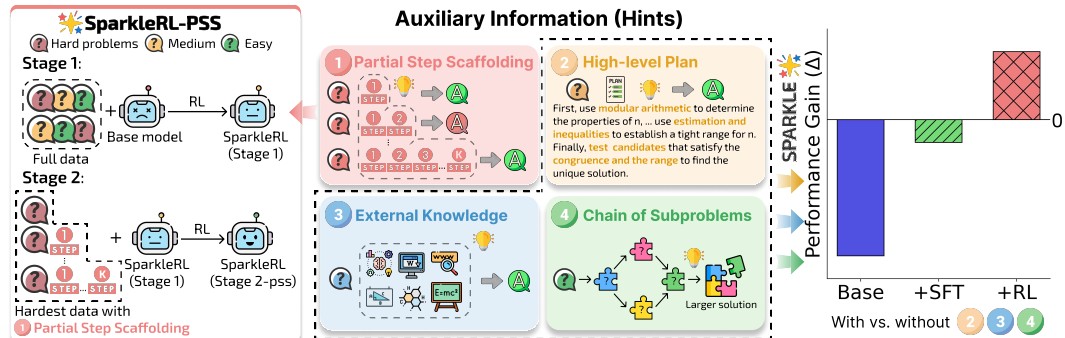

Figure 1: **Left:** SparkleRL-PSS, a two-stage curriculum-style RL training with *partial step scaffolding*—the hardest problems are revisited with auxiliary step-level hints to guide multi-stage learning. **Middle:** Four types of auxiliary information (hints) used in this work: (1)*Partial Step Scaffolding* (used in Stage 2 RL), (2) *High-level Plan*, (3) *External Knowledge*, and (4) *Chain of Subproblems* (2-4 used in the SPARKLE analysis framework). **Right:** Net performance gains/losses when models are evaluated with vs. without hints (2–4). While all models struggle on compositional subproblems, RL-tuned models exhibit the greatest flexibility in leveraging auxiliary information—whereas base performance drops sharply and SFT models show limited benefit.

To enable such analysis, we introduce SPARKLE, a fine-grained analysis framework that examines key elements hypothesized to benefit from RL training. Specifically, we evaluate pre- and post-RL tuned models along three dimensions: (1) plan-following and execution; (2) knowledge use; and (3) problem decomposition. Our framework enables fine-grained assessment of reasoning behaviors, revealing not just where RL enhances performance but also its limitations across different reasoning components.

Existing benchmarks, which consist of problems, ground-truth answers, and occasionally reasoning traces, are insufficient for the fine-grained analysis we perform. For example, testing a model's ability to follow a given plan requires access to planning annotations. To instantiate SPARKLE on mathematical reasoning, we augment math datasets with planning skeletons, requisite knowledge annotations, and candidate problem decompositions. A concrete example is shown in Figure 2. This approach produces novel insights: for example, we observe that ***giving base and SFT models access to human-crafted, correct plans can surprisingly degrade performance, while RL-tuned models are more robust to these, but the plan is better provided as high-level.*** This suggests that part of the benefits of RL are the flexibility to use multiple plausible plans.

We also investigate a second form of fine-grained analysis, related to *problem difficulty*. Prior work has observed that RL often fails to exploit hard problems effectively, as these examples rarely yield positive reward signals [49]. This has motivated filtering strategies that remove such problems—but this wastes valuable training signal. Instead, we analyze problem difficulty and use the resulting distinctions to ***develop a new multi-stage RL pipeline***(SparkleRL-PSS) that exploits data of varied difficulties. In the first stage, we perform RL on a broad set of diverse math problems, analogous to Guo *et al.* [15]. In the second, we fine-tune the model further on identified hard problems. To further help guide the model on these challenging cases, we give it access to partial solution augmentations or hints without further data generation. This two-stage setup is designed to first give the model a strong general reasoning boost, then hone its skills on the trickiest examples.

Together, SPARKLE and SparkleRL-PSS offer a comprehensive view of how RL shapes model behavior both analytically and algorithmically. They show that RL enhances flexibility in plan following and knowledge integration; however, performance degrades when models are forced to follow concrete, human-crafted plans. Instead, RL-tuned models perform best when guided by high-level plans that align with their internal reasoning dynamics, although robustness in solving chained subproblems remains limited. These findings highlight concrete directions for developing RL pipelines that are more adaptive, data-efficient, and interpretable for reasoning tasks. Our contributions include:

- We introduce SPARKLE, a novel analysis framework to systematically evaluate plan following, knowledge utilization, and subproblem solving in LLM reasoning.

**Problem:**

One of Euler's conjectures was disproved in the 1960s by three American mathematicians when they showed there was a positive integer such that

$$133^5 + 110^5 + 84^5 + 27^5 = n^5$$

Find the value of $n$.

**Planning:**

**Step 1:** Analyze the modular properties of both sides to determine possible congruences for n modulo small primes.

**Step 2:** Apply Fermat's Little Theorem to deduce the congruences of n modulo 2, 3, and 5.

**Step 3:** Use the Chinese Remainder Theorem to combine these congruences and find a specific congruence modulo 30.

**Step 4:** Generate candidate values for n that satisfy this congruence.

**Step 5:** Estimate upper bounds for n by comparing the sum on the left-hand side to potential values of $n^5$, using inequalities.

**Step 6:** Test specific values of n within the narrowed range to identify the value of n that satisfies the original equation.

**Knowledge:**

**Fact:** Euler's sum of powers conjecture posited that at least n nth powers are needed to sum to another nth power.

**Definitions:** 1) Modular arithmetic involves integers wrapping around upon reaching a certain value, the modulus; 2) The Chinese Remainder Theorem provides a way to solve systems of simultaneous congruences with pairwise coprime moduli.

**Theorems:** 1) Fermat's Little Theorem: If p is a prime number and a is an integer not divisible by p, then $a^{p-1} \equiv 1 \pmod{p}$; 2) Chinese Remainder Theorem: If one knows the remainders of the division of an integer n by several pairwise coprime integers, then one can determine uniquely the remainder of the division of n by the product of these integers.

**Subproblems:**

**Q1:** What remainder patterns emerge when $n^5$ is divided by the primes 2, 3, and 5?

**Q2:** Based on the result from Q1, what is n modulo 30?

**Q3:** What are values for n greater than 133 that satisfy n ≡ 24 mod 30?

**Q4:** What is an upper bound for n by comparing the sum $133^5 + 110^5 + 84^5 + 27^5$ to $n^5$?

**Q5:** What is the value of n that satisfies all previous conditions and the original equation?

Figure 2: Illustration of the SPARKLE framework's three-dimensional analysis approach. For each problem (***top***), we construct three complementary components: a high-level planning skeleton (***left***) capturing the overall solution strategy, relevant knowledge (***middle***) required for reasoning, and a sequence of interconnected subproblems (***right***) that decompose the solution process. The augmented benchmark enables a fine-grained understanding of reasoning capabilities and failure modes in LLMs.

- We construct SPARKLE benchmark, augmented with planning skeletons, knowledge information and subproblem chains to support comprehensive reasoning analysis.
- A simple yet effective multi-stage RL training approach with partial step scaffolding (SparkleRL-PSS) that reuses existing hard problems without additional data generation.
- We present comprehensive empirical findings that reveal which aspects of reasoning are most enhanced by RL (*e.g.*, flexibility in plan following and integrating knowledge into its reasoning processes), which remain challenging (*e.g.*, robustness in solving subproblems), and the conditions under which RL provides the greatest benefits.

## 2 SPARKLE: A Three-Axis Framework for LLM Reasoning Evaluation

To precisely analyze LLM reasoning, we introduce SPARKLE, a framework that decomposes reasoning along three axes: plan-following and execution, knowledge utilization, and subproblem decomposition (Section 2.1), inspired by classic research in cognitive science on human reasoning and problem solving [7, 30, 35, 44]. We also present a dataset constructed to instantiate this framework and support systematic evaluation (Section 2.2). Together, the framework and dataset enable fine-grained, interpretable analysis of key reasoning competencies in LLMs (Figure 2).

### 2.1 SPARKLE Analysis Framework Overview

**Axis 1: planning and execution.** When LLMs fail to solve challenging mathematical problems, is the cause not knowing *what* to do—or an *inability to carry out the steps*? We evaluate models on problems both 1) *with* and 2) *without* accompanying planning skeletons. The plan outlines the major steps needed but omits the details that must be carried out by the model. In the former case, the planning sketch alleviates the planning burden, allowing us to isolate and assess the model's execution capabilities (see a full example in Appendix B). By comparing performance under these conditions, we can better understand whether RL fine-tuning primarily enhances strategic planning, step-by-step execution, or both components of the reasoning process.

**Axis 2: knowledge utilization.** Mathematical reasoning relies on access to factual knowledge—including definitions, theorems, and formulas—and the ability to apply this knowledge: knowledge defines the premises, while reasoning describes the logical operations performed on those premises. Performance improvements from RL fine-tuning may stem from enhanced deductive reasoning abilities, improved knowledge utilization, or a combination of both.

Inspired by prior works on transparent logical reasoning [6], our second evaluation axis addresses this ambiguity by separating between knowledge retrieval and reasoning processes. Concretely, knowledge in our setting refres to the collection of relevant facts, definitions, theorems, and lemmas necessary for solving the problem, while reasoning encompasses the logical operations that manipulate and apply this knowledge toward the answer. We systematically vary *knowledge availability* to separate these roles. In one condition, the model must retrieve all relevant concepts itself. In the other, we explicitly provide the necessary knowledge (*e.g.*, statements of Fermat's Little Theorem and the Chinese Remainder Theorem, as in the example in Figure 2). This design allows us to identify knowledge-related bottlenecks. A model that succeeds only when given knowledge has intact reasoning but incomplete recall or because the information lies outside its training data. A model that still fails despite having all the facts reveals limits in deductive ability.

**Axis 3: chain of subproblems.** Even when an LLM gives a correct final answer, it may contain flawed intermediate steps [51]. To uncover where reasoning breaks down, we decompose problems into a chain of subproblems and assess model performance incrementally. At each stage, the model is shown the original problem, the subproblems solved so far, and the current subproblem. For example, when answering Q3 (Figure 2), the model is shown the full prompt and the answers to Q1 and Q2. Crucially, unlike the structured planning sketch in Axis 1, these subproblems are *not prescriptive instructions*. Instead, they act as checkpoints—smaller, self-contained problems that are individually solvable but collectively build toward the full solution. They provide no hints about what *method* to use, only what *question* to answer. This framing allows us to identify the precise step where the model's reasoning fails, offering a fine-grained error analysis.

**Remarks.** We focus on these three axes as they capture core aspects of reasoning that can be systematically and quantitatively evaluated (Section 5). These dimensions are not strictly orthogonal—for example, retrieved knowledge can inform planning—but together they offer a practical and interpretable framework for analyzing the impact of RL on reasoning behavior.

## 2.2 SPARKLE Benchmark Construction and Validation

Extant benchmarks—typically consisting of problems, answers, and reasoning traces—lack the components needed to study RL along the three axes we have proposed. To address this, we augment popular reasoning evaluation sets to produce the SPARKLE benchmark.

**Pipeline overview.** We construct our benchmark through a unified annotation pipeline that supports all three evaluation axes. For each problem, we begin with its ground-truth solution and prompt a high-capacity agent with access to the Internet (*e.g.*, GPT-4.1 [36]) to: (1) extract a planning skeleton summarizing key reasoning steps, (2) decompose the problem into a sequence of well-defined subproblems with answers, and (3) identify relevant knowledge components (facts, definitions, theorems, lemmas). The model is instructed to retrieve knowledge from reliable sources on the Internet when necessary. To ensure annotation quality across all three dimensions, we employ a second verification agent (*e.g.*, GPT-4.1) that checks the outputs for correctness, coherence, completeness, and pedagogical soundness. If any aspect fails, the annotations are regenerated. Finally, expert validation is conducted by graduate students with advanced mathematics background to ensure that the annotations faithfully capture the underlying reasoning and required knowledge.

**SPARKLE benchmark statistics.** SPARKLE is created from diverse mathematical problem benchmarks including AIME24 [32], AMC23 [31], MATH500 [17], GSM8K [5], and OlympiadBench [16] (test splits). Each problem is augmented with planning information derived from ground-truth reasoning traces, relevant knowledge components, and a sequence of subproblems curated via the pipeline described above. An example of the augmented problem is illustrated in Figure 2. We also annotate the difficulty level using AoPS Competition Ratings [1] and mathematical domain (*e.g.,* linear algebra, geometry, number theory). The resulting SPARKLE benchmark contains 2,564 open-ended questions spanning 10 difficulty levels and 9 domains. More details are provided in Appendix B.

# 3 Problem Difficulty and Its Implications on Reinforcement Learning

The SPARKLE framework enabled us to conduct a fine-grained evaluation of how reinforcement learning affects the reasoning capabilities of LLMs. This evaluation, however, did not examine *problem difficulty*. We now tackle this axis, again seeking insights into RL behavior.

A prominent belief is that performing RL on problems that are *too challenging* (e.g., beyond current frontier models, or too complex for low-capacity models) is not useful because models are unlikely to obtain any reward. Such samples are filtered out—at the cost of reducing our dataset size. We study this phenomenon, asking: *Can difficult problems still contribute meaningfully to learning?* To perform this study, we use two training setups:

**RL from base LLMs.** In the first setup, we fine-tune a base LLM using RL on mathematical problems. This setup mirrors standard approaches used in recent RLHF-style training pipelines where the reward is derived from correctness or other problem-specific heuristics. This setup serves as our baseline, offering insights into how general-purpose (single-stage) RL affects reasoning across a wide range of problem difficulties. In particular, we adopt Group Relative Policy Optimization (GRPO) [39], a variant of Proximal Policy Optimization (PPO) [38]. It has demonstrated remarkable performance on common benchmarks [15, 29] and is more computationally efficient than PPO. More details about GRPO are provided in Appendix C.

**Multi-stage RL from base LLMs.** To further probe how RL shapes reasoning under different conditions, focusing on varying difficulty, we introduce a second, more structured setup that aligns with curriculum learning principles [34][2]. In this multi-stage variant, we continue RL fine-tuning from the first-stage model checkpoint, on a subset of difficult problems selected from the full training set. This stage is designed to further enhance the model's ability by learning from challenging samples. Within this setup, we explore three curriculum variants: (1) **Mixed Difficulty** uses a random mixture of easy and hard problems to maintain exposure diversity; (2) **Hard-Only** restricts training to difficult problems, concentrating learning on high-complexity cases; and (3) **Hard-Augmented** (ours; SparkleRL-PSS) introduces partial solution scaffolding—such as intermediate steps or hints—to help the model navigate complex reasoning paths more effectively. These controlled variants allow us to assess how RL interacts with problem difficulty and solution augmentation. Additionally, we study how difficulty interacts with the planning, execution, and knowledge axes defined by our SPARKLE framework.

# 4 Experimental Setup

Next we provide the detail for the experiments we perform using the evaluation principles and approaches from Sections 2 and 3. First, to remove the potentially confounding effects of supervised fine-tuning (SFT), we apply RL directly to base pretrained LLMs. This complements prior studies on the impacts of SFT and the interplay of SFT and RL [4, 52].

**Reward design.** We use a rule-based reward following Guo *et al.* [15], which also mitigates the reward hacking problem of using a reward model [9, 10, 12]. We evaluate both answer correctness and solution format using the formula below:

$$R(\hat{y}, y) = \begin{cases} 2, & \text{if } \texttt{answer\_correct}(\hat{y}, y) \land \texttt{format\_correct}(\hat{y}) \\ 1, & \text{if } \texttt{answer\_correct}(\hat{y}, y) \land \neg\texttt{format\_correct}(\hat{y}) \\ -1, & \text{otherwise} \end{cases}$$

where $\hat{y}$ represents the model's generated answer, $y$ is the reference answer, $\texttt{answer\_correct}(\cdot)$ evaluates numerical equivalence, and $\texttt{format\_correct}(\cdot)$ assesses adherence to expected answer format. This encourages the model to answer correctly with encouraged format correctness.

**Training details.** For Stage 1, we use the training set from DeepScaleR-Preview [29], which contains 40K math questions spanning a wide range from AIME (1984-2023), AMC (pre-2023), MATH [17], Still [42], and Omni-MATH [11]. Our SparkleRL-Stage 1 model is trained on these 40K problems using GRPO. To curate the training set for Stage 2, we first identify 6.5K most challenging problems

---

[2]While difficulty-based sampling can be integrated into single-stage GRPO training by rejecting samples that are too easy or difficult based on their estimated *advantages* as done in DAPO [53], we opt for a two-stage setup to disentangle general RL effects from curriculum-driven improvements.

| Model | AIME24 | AMC23 | MATH500 | GSM8K | OlympiadBench | Avg. |
|---|---|---|---|---|---|---|
| Qwen-2.5-Math-7B-Base | 16.67 | 42.50 | 44.03 | 42.53 | 28.65 | 35.23 |
| SparkleRL-Stage 1 | $46.67_{\uparrow 30.00}$ | $67.50_{\uparrow 25.00}$ | $80.00_{\uparrow 35.97}$ | $91.77_{\uparrow 49.24}$ | $39.11_{\uparrow 10.46}$ | 65.01 |
| SparkleRL-Stage 2-hard | $41.67_{\uparrow 25.00}$ | $65.94_{\uparrow 23.44}$ | $80.50_{\uparrow 36.47}$ | $92.45_{\uparrow 49.92}$ | $37.39_{\uparrow 8.74}$ | 63.59 |
| SparkleRL-Stage 2-mix | $40.00_{\uparrow 23.33}$ | $63.44_{\uparrow 20.94}$ | $80.78_{\uparrow 36.75}$ | $\mathbf{92.52}_{\uparrow 49.99}$ | $38.85_{\uparrow 10.20}$ | 63.12 |
| SparkleRL-Stage 2-pss | $\mathbf{50.42}_{\uparrow 33.75}$ | $\mathbf{71.25}_{\uparrow 28.75}$ | $\mathbf{81.00}_{\uparrow 36.97}$ | $92.38_{\uparrow 49.85}$ | $\mathbf{40.11}_{\uparrow 11.46}$ | $\mathbf{67.03}$ |

Table 1: Performance comparison of Qwen-2.5-Math-7B and tuned models from multi-stage RL. We report results of metric Avg@8. We **bold** the best results.

that the best Stage 1 model fails to solve after 20 attempts. We then validate this subset using a GPT-4.1-based Web Agent [36], followed by human verification to further filter out items with flawed solutions. This results in a curated set of 5.7K difficult problems. For problems lacking reasoning traces, we adopt reference solutions from NuminaMath [27].

For Stage 2, we initialize from SparkleRL-Stage 1 and explore three fine-tuning variants. SparkleRL-Stage 2-mix is trained on a mixture of easy and hard problems. SparkleRL-Stage 2-hard is trained on the 5.7K most difficult problems identified from Stage 1. SparkleRL-Stage 2-pss is trained on the same set of difficult problems, but with partial solution augmentation: each reasoning trace is divided into four semantic chunks, and for each problem, we construct multiple examples by providing between 0 and 4 chunks as additional input context (Figure 1, left).[3] The model is then prompted to complete the reasoning and arrive at the final answer.

# 5 Results

We present our main findings. Our evaluation focuses on both high-level performance outcomes and a fine-grained analysis of reasoning capabilities, guided by the following key questions:

- **Multi-Stage RL and Role of Problem Difficulty** (Section 5.1): How effective is multi-stage RL at improving reasoning performance? How do problem difficulties impact RL? We show that appropriately structured hard problems can provide additional benefits.
- **Sample Efficiency** (Section 5.2): Does RL improve the model's ability to solve problems with fewer samples? We show that RL-tuned models achieve higher performance at lower attempts compared to base models.
- **Plan Following and Execution** (Section 5.3): How does RL impact the ability to follow externally provided plans? We find that RL-tuned models demonstrate improved flexibility in plan following yet often perform better with self-generated planning strategies.
- **Knowledge Integration** (Section 5.4): Can RL enhance a model's ability to use external knowledge? Our results reveal that RL-tuned models show significant improvements when provided with supplementary knowledge, while base models struggle.
- **Subproblem Resolution** (Section 5.5): Does RL improve the model's ability to systematically solve decomposed problems? We observe that while RL substantially improves overall performance, all models still struggle with detailed subproblem resolution.
- **Scaling Reasoning Benefits with Task Difficulty** (Section 5.6): How do the benefits of knowledge and planning guidance vary with problem difficulty? We demonstrate that knowledge integration becomes increasingly valuable as problem difficulty rises.

## 5.1 Are Difficult Problems Still Valuable for RL training?

Table 1 presents a performance comparison between Qwen-2.5-Math-7B and several RL-tuned variants across five benchmarks, reporting Avg@8 scores. Stage-1 training on the full dataset establishes strong reasoning, achieving substantial gains across all benchmarks (an average of 29.78% improvement). For the second stage, we systematically investigate the impact of **problem difficulty** on RL training through three variants: training exclusively on hard problems, using a mixture of easy and hard problems, and employing hard problems augmented with partial solutions (as detailed in Sec. 3). The results reveal that while further training on harder

---

[3]The CoT traces used for partial-step scaffolding are of mixed quality and come directly from the existing datasets, so no additional data generation or external distillation is performed.

problems or mixed-difficulty problems improves performance on simpler tasks like GSM8K (difficulty level 1/10) and MATH500 (difficulty level 1.5/10), performance decreases on more challenging benchmarks. Contrary to prior work suggesting that GRPO cannot benefit from the hardest problems due to absent positive reward signals [49], we demonstrate that ***hard problems remain valuable when appropriately structured***. Specifically, while training solely on hard problems yields modest additional gains, augmenting them with partial solution guidance proves most effective, consistently improving performance across all benchmarks by enabling models to navigate complex reasoning paths more systematically.[4] This approach yields an average improvement of 2.02% over Stage 1 and a new peak of 50.42% on the most challenging task AIME24—performance comparable to SoTA pure RL-tuned 32B models (50% on AIME24 [53]). Statistical significance tests confirming these improvements are provided in Appendix H.

## 5.2 Does RL Improve Sample Efficiency?

Figure 3 presents pass@k results across models. In the following sections, we use SparkleRL-Stage 2-pss as the representative model for Stage 2 since it demonstrates the best performance among the three variants. We observe two key patterns: (1) multi-stage RL with partial solution augmentation (SparkleRL-Stage 2-pss) consistently outperforms single-stage training, achieving higher performance at lower k values compared to both Stage 1 and the base model; and (2) as k increases, the performance gap between all three models gradually narrows. However, improved sampling efficiency alone cannot fully explain the observed gains. These gains may reflect fundamental shifts in model behavior. What is really happening under the hood? Next, we dig deeper to uncover the specific capabilities enhanced by RL by dissecting reasoning through SPARKLE.

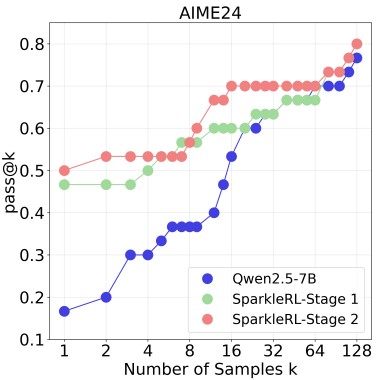

Figure 3: Pass@k comparison between Qwen-2.5-Math-7B, SparkleRL-Stage 1, and SparkleRL-Stage 2.

## 5.3 How does RL Impact Plan Following?

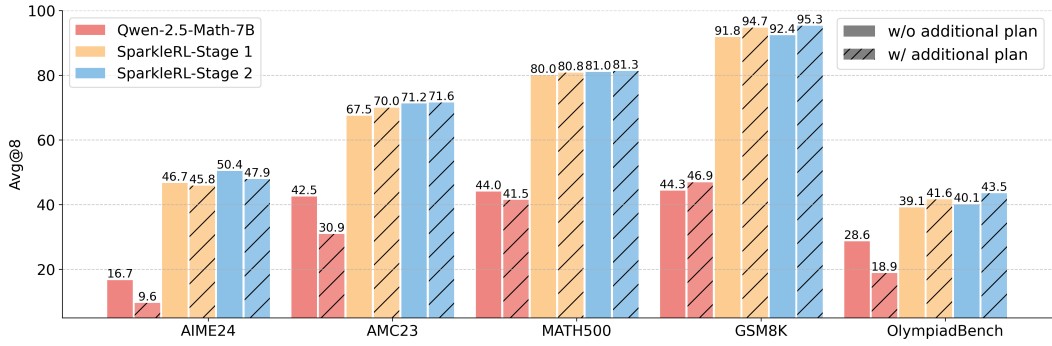

Figure 4: Performance comparison of Qwen-2.5-Math-7B, SparkleRL-Stage 1, and SparkleRL-Stage 2 with and without additional planning information. RL-tuned models (Stage 1 and Stage 2) maintain performance with planning guidance, while the base model shows performance degradation in four out of five benchmarks when provided with plans. AIME24 exhibits the most pronounced effect where even RL-tuned models perform better without externally imposed plans.

**RL-tuned models demonstrate improved flexibility in plan following and execution.** Planning is integral for problem-solving. Surprisingly, we find that *a valid plan, derived from human solutions, is not necessarily a good plan for models to execute.* Without externally imposed constraints, models can generate more reliable planning structures, such as functional Python code. Using predefined planning templates, while seemingly advantageous, paradoxically increases the likelihood of overlooking

---

[4]It is crucial to provide complete reasoning chunks rather than only the initial steps, as partial augmentations may limit the benefit of guided reasoning.

corner cases, resulting in incorrect final answers (Table 3). Figure 4 quantifies these effects across benchmarks. The base model's performance drops in every task except GSM8K. For elementary tasks like GSM8K (difficulty 1/10), the base model already possesses basic planning ability and benefits from explicit step-by-step instructions. For example, when the base model fails by attempting simultaneous calculations, it succeeds once guided to decompose the steps.

RL-tuned models, however, display stronger plan-following flexibility. Their performance remains stable or improves slightly with additional plans—except on the most difficult task, AIME24, both Stage 1 and Stage 2 see performance drop (*e.g.*, Stage 2 decreases from 50.4% to 47.9%). Importantly, RL-tuned models consistently perform best when allowed to develop their own planning strategies rather than following human-derived ones. This suggests that RL fosters internal strategies aligned with the model's reasoning dynamics, while externally imposed plans may conflict with the heuristics learned during training.

Additional SFT experiments (Appendix G) confirm that RL enhances plan-following flexibility beyond instruction following and Appendix E reports additional results for the 32B model.

## 5.4 Does RL Improve Knowledge Utilization?

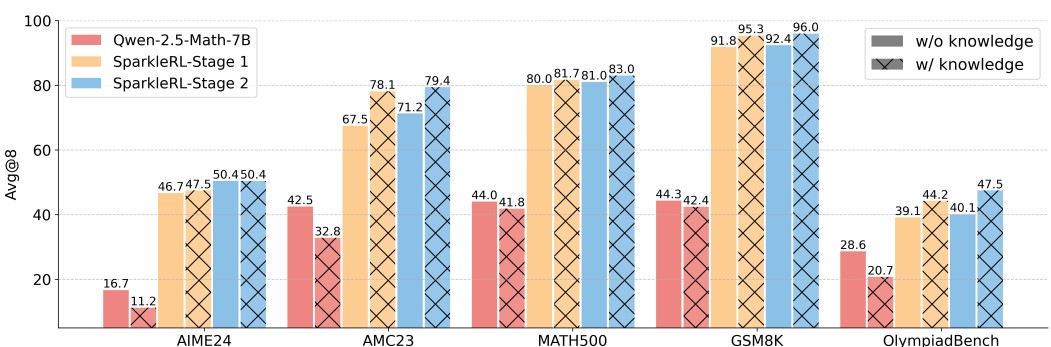

Figure 5: Comparison of Qwen-2.5-Math-7B, SparkleRL-Stage 1, and SparkleRL-Stage 2 with and without knowledge information. The base model shows consistent performance degradation (avg. 5.4% decrease) when provided with external knowledge, RL-tuned models show significant performance improvements (4.3% and 4.2% avg. gains for Stage 1 and Stage 2 models, respectively).

**RL-tuned models exhibit enhanced knowledge integration capabilities despite inherent knowledge limitations.** Figure 5 compares the performance of Qwen-2.5-Math-7B, SparkleRL-Stage 1, SparkleRL-Stage 2-pss with and without access to supplementary knowledge. The base model's performance consistently declines when given external knowledge (average drop of 5.4% across five tasks), indicating fundamental limitations in its ability to incorporate external information into its reasoning process efficiently. In contrast, both RL-tuned variants show substantial gains when provided with the same knowledge—averaging improvements of 4.3% (Stage 1) and 4.2% (Stage 2). This is a critical distinction between base and RL-tuned models and suggests that while these RL-tuned models still exhibit knowledge limitations, they have developed robust mechanisms for integrating new information during inference. A practical takeaway: rather than relying solely on continued RL fine-tuning—which may risk catastrophic forgetting—***providing targeted external knowledge is a simple and effective way to enhance performance*** on knowledge-intensive tasks.

Appendix E further shows that for the 32B model, knowledge augmentation provides larger gains than planning for RL-tuned variants.

## 5.5 Can RL Solve Decomposed Hard Problems?

**RL-tuned models still struggle with detailed subproblem resolution.** To test whether RL improves systematic problem decomposition, we compare performance on full problems versus their constituent subproblems. Figure 6 shows results for Qwen-2.5-Math-7B, SparkleRL-Stage 1 and SparkleRL-Stage 2-pss on original problems vs. their ability to solve all subproblems of those same problems.

Let $P$ be a problem which can be decomposed into $K$ subproblems $\{s_1, s_2, \ldots, s_K\}$. The subproblem success rate $\text{SSR}(P)$ is defined as: 1 if the model correctly solves all subproblems

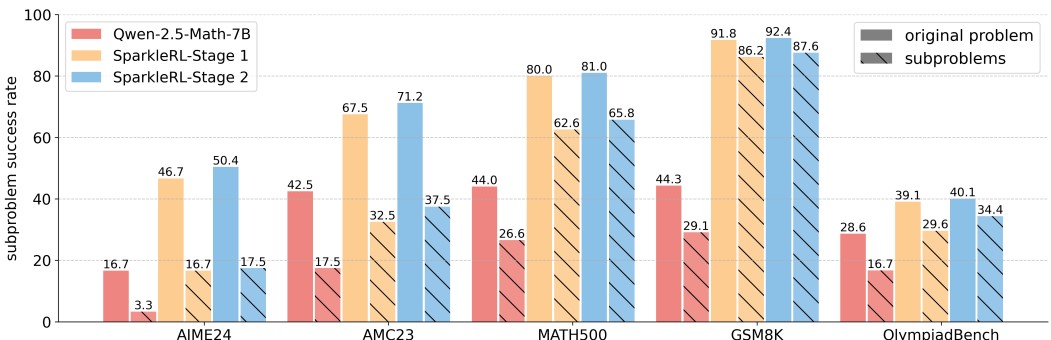

Figure 6: Comparison of Qwen-2.5-Math-7B, SparkleRL-Stage 1, and SparkleRL-Stage 2 with original problems and average subproblem success rate. Results show a consistent performance gap between solving complete problems and successfully addressing all constituent subproblems.

and 0 otherwise. For a set of $N$ problems $\{P_1, P_2, ..., P_N\}$, the average subproblem success rate is $\bar{\text{SSR}} = \frac{1}{N}\sum_{i=1}^{N}\text{SSR}(P_i)$. Across all tasks, both base and RL-tuned models show a large gap between original accuracy and $\bar{\text{SSR}}$. On AIME24—the hardest benchmark—Qwen-2.5-Math-7B reaches 16.7% accuracy on full problems but only 3.3% on subproblems; SparkleRL-Stage 2-pss achieves 50.4% on full problems but just 17.5% on subproblems. Viewed alongside the planning results in Figure 4, this suggests that *RL-tuned models benefit from high-level planning guidance but remain weak at detailed decomposition and resolution*. Even when subproblems appear simpler in isolation, the difficulty lies in solving every component consistently. Overall, current RL methods *favor autonomous high-level strategies that align with a model's internal dynamics*, while effective decomposed problem solving would likely require *new methods tailored to ensure consistency across subproblems*.

## 5.6 A Closer Look at Knowledge and Planning by Difficulty Level

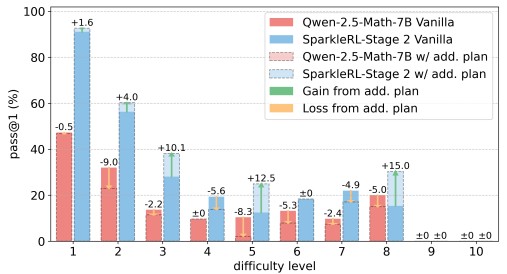

(a) With and without additional planning information. RL-tuned models maintain stable across difficulties; base model degrades as difficulty increases.

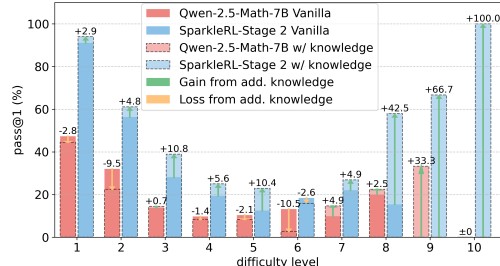

(b) With and without knowledge information. Knowledge augmentation benefits RL-tuned models more, especially on harder problems.

Figure 7: Base model vs. RL-tuned model pass@1 by difficulty level.

**Knowledge integration becomes increasingly valuable as problem difficulty rises, while planning benefits remain relatively constant.** Figure 7a and 7b show how planning and knowledge augmentation varies with problem difficulty. Across levels 1–8 (where sample sizes are reliable), supplementary knowledge consistently outperforms planning, and this advantage grows as tasks become harder. For example, at level 7 (41 problems), knowledge augmentation improves RL-tuned performance by +4.9% while planning reduces accuracy by -4.9%; at level 8 (40 problems), knowledge yields a dramatic +42.5% gain compared to +15% from planning. Weighted across levels, knowledge provides an average improvement of +4.53%, compared to +2.50% from planning.

For RL-tuned models, these results highlight that external knowledge is a key driver of performance on complex problems, while planning contributes smaller and less consistent gains. In contrast, base models show the opposite pattern: planning often harms performance as difficulty increases, and knowledge yields only modest improvements. This asymmetry indicates that base models

are less capable of leveraging auxiliary information effectively, whereas RL-tuned models develop mechanisms to integrate knowledge in ways that meaningfully enhance performance.

Together, these findings reinforce our broader thesis: *RL fundamentally reshapes how models process and integrate auxiliary information.* As difficulty increases, this ability becomes a *key differentiator* between base and RL-tuned models, particularly in knowledge-intensive reasoning tasks.

## 6 Related Work

**Understanding Reinforcement Learning for LLM Reasoning.** Gradient-based policy optimization algorithms [15, 38, 39] with verifiable objectives have shown remarkable performance on reasoning-intensive tasks [2, 37, 47]. Curriculum-based methods [3, 24, 41, 57], such as difficulty-aware sampling, have been used to improve SFT and RL training for LLMs [19, 26, 48, 53]. Despite these advances, the mechanism by which RL shapes reasoning remains an open question. Previous studies have explored the interplay between SFT and RL in text-based [52] and visual environments [4], but they only involve a single-stage RL and do not dissect RL's effects beyond overall accuracy. Yue *et al.* [54] argued that RL-tuned models mainly reweight reasoning paths rather than creating new capabilities. Several recent surveys provide broader overviews of this evolving area [21, 33, 50, 56]. Our work moves beyond accuracy metrics for LLM reasoning under multi-stage RL.

**Diverse Aspects of LLM Reasoning.** Reasoning in LLMs has attracted significant attention in recent years [20, 22]. At its core, reasoning is a cognitive process that integrates evidence, arguments, and logic to reach conclusions or judgments. Research in cognitive science [7, 30, 35, 44] highlights the interplay of knowledge, planning, and problem decomposition as fundamental components of human problem solving. In the context of LLMs, knowledge retrieval and utilization [14, 25, 43], subproblem decomposition [8, 23, 46, 58], and planning [45] have also been explored individually. However, it remains largely underexplored how RL shapes these crucial dimensions—a critical gap we address in this work.

**Mathematical Reasoning Benchmarks.** Mathematical problem-solving has become a central testbed for evaluating LLM reasoning capabilities [11, 27, 29, 42]. While earlier benchmarks such as GSM8K [5] and MATH [18] target grade-school and competition-level mathematics, newer models perform strongly on these tasks, necessitating harder benchmarks such as AMC12 and AIME [32]. Recent models show impressive results: OpenAI-o3 model scored 91.6% in AIME2024, and DeepSeek-R1 [28] reached 97.3% on MATH500. Nevertheless, most of these benchmarks—despite their utility—provide only coarse-grained signals of reasoning ability with little insight into internal processes, motivating our more fine-grained diagnostic framework.

## 7 Discussion and Conclusion

We investigated *if and how* reinforcement learning shapes the reasoning capabilities of LLMs. To this end, we proposed SPARKLE, a fine-grained analytic framework that decomposes reasoning into plan following, knowledge integration, and subproblem solving. By augmenting existing mathematical reasoning benchmarks with human verification, we built the SPARKLE benchmark for detailed analysis. Our findings show that RL improves flexibility in plan following and knowledge utilization, yet compositional subproblem solving remains fragile. Interestingly, human-crafted plans can hinder RL-tuned models, which prefer autonomous, high-level strategies aligned with their internal reasoning. In contrast, lightweight external knowledge injection proves more beneficial, particularly for difficult tasks. We further introduce SparkleRL-PSS, a multi-stage RL pipeline that reuses hard problems with partial step scaffolding—avoiding new data generation while effectively guiding exploration.

Looking forward, we highlight two promising directions: (1) *Data perspective:* developing methods to make diverse and imperfect data to provide effective guiding signals for RL, while aligning models' intrinsic reasoning strategies; and (2) *Training perspective:* systematically incorporating auxiliary hints for difficult problems, such as high-level plans, modular knowledge, or subproblem cues, into RL training to induce richer and more adaptive reasoning behavior. Overall, our framework, method, and findings shed light on *if and how* RL contributes to reasoning, offering practical insights for developing more intelligent, data-efficient, and interpretable RL pipelines for LLMs.

## Acknowledgement

The authors would like to thank NeurIPS anonymous reviewers for their insightful feedback and helpful discussions. We thank Aditya Goyal for data verification. We are grateful for the support of the NSF under #2106707 and #2402833 and the Defense Advanced Research Projects Agency (DARPA Young Faculty Award).

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

# Appendix

# A   Limitations and Societal Impacts

**Limitations**   While our analysis offers a detailed empirical view of how multi-stage RL shapes reasoning across plan following and execution, knowledge use, and problem decomposition, several limitations remain. We typically focus on structured reasoning problems (*e.g.*, math), and it may require adaptation for domains with less structured and explicit decomposition. The dataset construction process, though expert-validated, depends on human annotation and may face scalability challenges. Lastly, our findings are empirical; developing theoretical tools to characterize internal reasoning strategies remains an important direction for future work.

**Societal Impacts**   Our work contributes tools and insights for building more transparent and interpretable reasoning models. By identifying how RL enhances specific reasoning behaviors, our framework can guide more targeted and efficient model development, especially in high-stakes domains such as education or science. While our datasets are math-focused and not privacy-sensitive, applying this methodology to broader domains will require careful attention to fairness and alignment. We hope this framework encourages more robust and trustworthy training practices for reasoning-capable LLMs.

# B   SPARKLE Dataset Details

Following Section 2.2, we present more details for the constructed SparkleRL datasets.

SPARKLE is created based on diverse mathematical problem benchmarks including AIME24 [32], AMC23 [31], MATH500 [17], GSM8K [5], and OlympiadBench [16] (test splits). Each problem is augmented with planning information derived from the groundtruth reasoning traces, relevant knowledge components, and a sequence of subproblems curated via the pipeline introduced in Section 2.2. We also annotate the difficulty level (template shown in Figure 8) and mathematical domain following [11] (*e.g.*, linear algebra, geometry, number theory). The resulting SPARKLE benchmark contains 2,564 open-ended questions spanning 10 difficulty levels and 9 domains.

## B.1   A full example of the SPARKLE dataset

We present a complete example of the SPARKLE benchmark in Figure 9. For the problem "One of Euler's conjectures...", we include its answer "144" and step-by-step solution "Taking the given equation modulo 2,3, and 5...". We also add its difficulty level of "4" (averaged from three GPT-4.1 ratings), the domain "Number Theory $\rightarrow$ Congruences", and a high-level solution plan "Step 1: Analyze the modular properties...", related knowledge "Fact: Euler's sum of powers conjecture posited...", and smaller subproblems "Q1: What remainder patterns emerge...". This enable a finer-grained evaluation of how reasoning models work and where they fail across different difficulty levels and domains.

---

**Template for Labeling Difficulty Level**

<system_role>
You are an expert grader for mathematics problems. Given a Problem and a Solution, estimate the problem's difficulty on a 1–10 scale according to the AoPS standard.
Below is the AoPS standard for difficulty estimation:
<requirements>
All levels are estimated and refer to averages. The following is a rough standard based on the USA tier system AMC 8 – AMC 10 – AMC 12 – AIME – USAMO/USAJMO – IMO, representing Middle School – Junior High – High School – Challenging High School – Olympiad levels. Other contests can be interpolated against this.
Notes: Multiple-choice tests like the AMC are rated as though they are free-response. Test-takers can use the answer choices as hints and therefore correctly answer more AMC questions than Mathcounts or AIME problems of similar difficulty. Some Olympiads are taken in two sessions, with two similarly difficult sets of questions numbered as one set. For these, the first half of the test (questions 1–3) is of similar difficulty to the second half (questions 4–6).
**Scale**
1: Problems strictly for beginners, on the easiest elementary or middle-school levels (MOEMS, MATHCOUNTS Chapter, AMC 8 1–20, AMC 10 1–10, AMC 12 1–5, and others that involve

---

standard techniques introduced up to the middle-school level); most traditional middle/high-school word problems.

2: For motivated beginners; harder questions from the previous categories (AMC 8 21–25, harder MATHCOUNTS States questions, AMC 10 11–20, AMC 12 5–15, AIME 1–3); traditional middle/high-school word problems with more complex problem solving.

3: Advanced-beginner problems that require more creative thinking (harder MATHCOUNTS Nationals questions, AMC 10 21–25, AMC 12 15–20, AIME 4–6).

4: Intermediate-level problems (AMC 12 21–25, AIME 7–9).

5: More difficult AIME problems (10–12); simple proof-based Olympiad-style problems (early JBMO questions, easiest USAJMO 1/4).

6: High-level AIME-style questions (13–15); introductory Olympiad-level questions (harder USAJMO 1/4 and easier USAJMO 2/5; easier USAMO and IMO 1/4).

7: Tougher Olympiad-level questions; may require more technical knowledge (harder USAJMO 2/5 and most USAJMO 3/6; extremely hard USAMO and IMO 1/4; easy–medium USAMO and IMO 2/5).

8: High-level Olympiad problems (medium–hard USAMO and IMO 2/5; easiest USAMO and IMO 3/6).

9: Expert Olympiad problems (average USAMO and IMO 3/6).

10: Historically hard problems, generally unsuitable for even very hard competitions (such as the IMO) due to being exceedingly tedious, long, and difficult (e.g., very few students worldwide are capable of solving them).

**Examples**

For reference, here are problems from each of the difficulty levels 1–10:

1: Jamie counted the number of edges of a cube, Jimmy counted the number of corners, and Judy counted the number of faces. They then added the three numbers. What was the resulting sum?

1: Let trapezoid $ABCD$ be such that $AB \parallel CD$. Additionally, $AC = AD = 5$, $CD = 6$, and $AB = 3$. Find $BC$.

1: How many integer values of $x$ satisfy $|x| < 3\pi$?

1: The whole number $N$ is divisible by 7. $N$ leaves a remainder of 1 when divided by 2, 3, 4, or 5. What is the smallest possible value of $N$?

1: The value of a two-digit number is 10 times the sum of its digits. The units digit is 1 more than twice the tens digit. Find the number.

1: The coordinates of $\triangle ABC$ are $A(5, 7)$, $B(11, 7)$, and $C(3, y)$ with $y > 7$. The area of $\triangle ABC$ is 12. What is $y$?

1: How many different three-digit whole numbers can be formed using the digits 4, 7, and 9, assuming that no digit is repeated?

1.5: A number is called *flippy* if its digits alternate between two distinct digits. For example, 2020 and 37373 are flippy, but 3883 and 123123 are not. How many five-digit flippy numbers are divisible by 15?

1.5: A rectangular box has integer side lengths in the ratio $1 : 3 : 4$. Which of the following could be the volume of the box?

1.5: Two lines with slopes $\frac{1}{4}$ and $\frac{5}{4}$ intersect at $(1, 1)$. What is the area of the triangle formed by these two lines and the vertical line $x = 5$?

2: A fair six-sided die is repeatedly rolled until an odd number appears. What is the probability that every even number appears at least once before the first occurrence of an odd number?

2: A small airplane has 4 rows of seats with 3 seats in each row. Eight passengers have boarded the plane and are distributed randomly among the seats. A married couple is next to board. What is the probability there will be two adjacent seats in the same row for the couple?

2: Suppose that $\frac{2009}{2014} + \frac{2019}{n} = \frac{a}{b}$, where $a$, $b$, and $n$ are positive integers with $\frac{a}{b}$ in lowest terms. What is the sum of the digits of the smallest positive integer $n$ for which $a$ is a multiple of 1004?

2.5: $A$, $B$, $C$ are three piles of rocks. The mean weight of the rocks in $A$ is 40 pounds, in $B$ is 50 pounds, in the combined piles $A$ and $B$ is 43 pounds, and in the combined piles $A$ and $C$ is 44 pounds. What is the greatest possible integer value for the mean, in pounds, of the rocks in the combined piles $B$ and $C$?

2.5: For some positive integer $k$, the repeating base-$k$ representation of the (base-ten) fraction $\frac{7}{51}$ is $0.\overline{23}_k = 0.232323\ldots_k$. What is $k$?

3: Triangle $ABC$ with $AB = 50$ and $AC = 10$ has area 120. Let $D$ be the midpoint of $\overline{AB}$, and let $E$ be the midpoint of $\overline{AC}$. The angle bisector of $\angle BAC$ intersects $\overline{DE}$ and $\overline{BC}$ at $F$ and $G$, respectively. What is the area of quadrilateral $FDBG$?

3: Wayne has 3 green buckets, 3 red buckets, 3 blue buckets, and 3 yellow buckets. He randomly distributes 4 hockey pucks among the green buckets, with each puck equally likely to be put in each bucket. Similarly, he distributes 3 pucks among the red buckets, 2 among the blue buckets, and 1 among the yellow buckets. Once he is finished, what is the probability that a green bucket contains more pucks than each of the other 11 buckets?

3: An object in the plane moves from one lattice point to another. At each step, the object may move one unit to the right, left, up, or down. If the object starts at the origin and takes a ten-step path, how many different points could be the final point?

3: Consider the integer

$$N = 9 + 99 + 999 + 9999 + \cdots + \underbrace{99\ldots99}_{321 \text{ digits}}.$$

Find the sum of the digits of $N$.

3: Let $\triangle LMN$ have side lengths $LM = 15$, $MN = 14$, and $NL = 13$. Let the angle bisector of $\angle MLN$ meet the circumcircle of $\triangle LMN$ at a point $T \neq L$. Determine the area of $\triangle LMT$.

3.5: Find all three-digit numbers $abc$ (with $a \neq 0$) such that $a^2 + b^2 + c^2$ is a divisor of 26.

3.5: Consider polynomials $P(x)$ of degree at most 3, each of whose coefficients is in $\{0, 1, 2, 3, 4, 5, 6, 7, 8, 9\}$. How many such polynomials satisfy $P(-1) = -9$?

3.5: Find the number of integer values of $k$ in the closed interval $[-500, 500]$ for which the equation $\log(kx) = 2\log(x + 2)$ has exactly one real solution.

3.5: In a drawer there are at most 2009 balls, some white and the rest blue, which are randomly distributed. If two balls are taken at the same time, the probability that the balls are both blue or both white is $\frac{1}{2}$. Determine the maximum possible number of white balls in the drawer such that the probability statement is true.

3.5: Find three isosceles triangles, no two of which are congruent, with integer sides, such that each triangle's area is equal to six times its perimeter.

4: Define a sequence recursively by $x_0 = 5$ and

$$x_{n+1} = \frac{x_n^2 + 5x_n + 4}{x_n + 6}$$

for all nonnegative integers $n$. Let $m$ be the least positive integer such that

$$x_m \leq 4 + \frac{1}{2^{20}}.$$

In which of the following intervals does $m$ lie?

4: An ant makes a sequence of moves on a cube where a move consists of walking from one vertex to an adjacent vertex along an edge. Initially the ant is at a vertex of the bottom face and chooses one of the three adjacent vertices as its first move. For all moves after the first, the ant does not return to its previous vertex but chooses to move to one of the other two adjacent vertices. All choices are equally likely. The probability that after exactly 8 moves the ant is at a vertex of the top face is $\frac{m}{n}$, where $m$ and $n$ are relatively prime positive integers. Find $m + n$.

4: Find all real numbers $a, b, c, d$ such that

$$\begin{cases} a + b + c + d = 20, \\ ab + ac + ad + bc + bd + cd = 150. \end{cases}$$

**(A)** $[9, 26]$  **(B)** $[27, 80]$  **(C)** $[81, 242]$  **(D)** $[243, 728]$  **(E)** $[729, \infty)$

4: The vertices of an equilateral triangle lie on the hyperbola $xy = 1$, and a vertex of this hyperbola is the centroid of the triangle. What is the square of the area of the triangle?

4: Isosceles trapezoid $ABCD$ has parallel sides $AD$ and $BC$, with $BC < AD$ and $AB = CD$. There is a point $P$ in the plane such that $PA = 1$, $PB = 2$, $PC = 3$, and $PD = 4$. What is $BC/AD$?

4.5: Find, with proof, all positive integers $n$ for which $2^n + 12^n + 2011^n$ is a perfect square.

4.5: Find the minimum value of

$$f(x) = x^{2008} - 2x^{2007} + 3x^{2006} - 4x^{2005} + \cdots - 2006x^3 + 2007x^2 - 2008x + 2009$$

over all real numbers $x$.

4.5: Show that the equation $a^2b^2 + b^2c^2 + 3b^2 - c^2 - a^2 = 2005$ has no integer solutions.

5: Triangle $ABC$ has side lengths $AB = 7$, $BC = 8$, and $CA = 9$. Circle $\omega_1$ passes through $B$ and is tangent to line $AC$ at $A$. Circle $\omega_2$ passes through $C$ and is tangent to line $AB$ at $A$. Let $K$ be the intersection of circles $\omega_1$ and $\omega_2$ not equal to $A$. Then $AK = \frac{m}{n}$, where $m$ and $n$ are relatively prime positive integers. Find $m + n$.

5: A pair of integers $(m, n)$ is called good if

$$m \mid (n^2 + n) \quad \text{and} \quad n \mid (m^2 + m).$$

Given relatively prime integers $a, b > 1$, prove that there exists a good pair $(m, n)$ with $a \mid m$ and $b \mid n$, but $a \nmid n$ and $b \nmid m$. 5: Let $ABCD$ be a convex quadrilateral with $\angle DAC = \angle BDC = 36°$, $\angle CBD = 18°$, and $\angle BAC = 72°$. The diagonals intersect at point $P$. Determine the measure of $\angle APD$.

5: Call a positive real number *groovy* if it can be written in the form $\sqrt{n} + \sqrt{n+1}$ for some positive integer $n$. Show that if $x$ is groovy, then for any positive integer $r$, the number $x^r$ is also groovy.

5: Find all prime numbers $p, q, r$ such that $\frac{p}{q} - \frac{4}{r+1} = 1$.

5: There are $a + b$ bowls arranged in a row, numbered 1 through $a + b$, where $a$ and $b$ are given positive integers. Initially, each of the first $a$ bowls contains an apple, and each of the last $b$ bowls contains a pear. A legal move consists of moving an apple from bowl $i$ to bowl $i + 1$ and a pear from bowl $j$ to bowl $j - 1$, provided that the difference $i - j$ is even. Multiple fruits may occupy the same bowl. The goal is to end with the first $b$ bowls each containing a pear and the last $a$ bowls each containing an apple. Show that this is possible if and only if the product $ab$ is even.

5: Solve the equation $3^x - 5^y = z^2$ in positive integers.

5: Find all triples $(a, b, c)$ of real numbers such that

$$a + b + c = \frac{1}{a} + \frac{1}{b} + \frac{1}{c}, \qquad a^2 + b^2 + c^2 = \frac{1}{a^2} + \frac{1}{b^2} + \frac{1}{c^2}.$$

5.5: Semicircle $\Gamma$ has diameter $\overline{AB}$ of length 14. Circle $\omega$ lies tangent to $\overline{AB}$ at a point $P$ and intersects $\Gamma$ at points $Q$ and $R$. If $QR = 3\sqrt{3}$ and $\angle QPR = 60°$, then the area of $\triangle PQR$ equals $\frac{a\sqrt{b}}{c}$, where $a$ and $c$ are relatively prime positive integers, and $b$ is a positive integer not divisible by the square of any prime. Find $a + b + c$. 5.5: Triangle $ABC$ has $\angle BAC = 60°$, $\angle CBA \leq 90°$, $BC = 1$, and $AC \geq AB$. Let $H$, $I$, and $O$ be the orthocenter, incenter, and circumcenter of $\triangle ABC$, respectively. Assume that the area of pentagon $BCOIH$ is maximized. What is $\angle CBA$?

6: Given an acute triangle $ABC$. The incircle of $\triangle ABC$ touches $BC, CA, AB$ at $D, E, F$, respectively. The angle bisector of $\angle A$ meets $DE$ and $DF$ at $K$ and $L$, respectively. Suppose $AA_1$ is an altitude of $\triangle ABC$, and let $M$ be the midpoint of $BC$. (a) Prove that $BK$ and $CL$ are perpendicular to the angle bisector of $\angle BAC$. (b) Show that $A_1KML$ is cyclic.

6: Let $ABCD$ be a convex quadrilateral. Let $I = AC \cap BD$, and let $E$, $H$, $F$, and $G$ lie on $AB$, $BC$, $CD$, and $DA$, respectively, such that $EF \cap GH = I$. If $M = EG \cap AC$ and $N = HF \cap AC$, show that

$$\frac{AM}{IM} \cdot \frac{IN}{CN} = \frac{IA}{IC}.$$

6: A $4 \times 4$ table is divided into 16 white unit square cells. Two cells are neighbors if they share a side. A move consists of choosing a cell and toggling the colors of its neighbors. After exactly $n$ moves all 16 cells are black. Find all possible values of $n$.

6: A magic $3 \times 5$ board can toggle its cells between black and white. Define a *pattern* to be an assignment of black or white to each of the 15 cells (so there are $2^{15}$ patterns total). Every day after Day 1, at the beginning of the day, the board creates a new pattern. However, the board always wants to be unique and will die if any two of its patterns are fewer than 3 cells different from each other. Furthermore, the board dies if it becomes all white. If the board begins with all cells black on Day 1, compute the maximum number of days it can stay alive.

6: Let $a, b, c$ be positive real numbers such that $a + b + c = 4\sqrt[3]{abc}$. Prove that

$$2(ab + bc + ca) + 4\min(a^2, b^2, c^2) \geq a^2 + b^2 + c^2.$$

6: Let $MN$ be a line parallel to side $BC$ of triangle $ABC$, with $M$ on $AB$ and $N$ on $AC$. The lines $BN$ and $CM$ meet at $P$. The circumcircles of triangles $BMP$ and $CNP$ meet again at $Q \neq P$. Prove that $\angle BAQ = \angle CAP$.

6: Let $\mathcal{P}$ be a convex $n$-gon with $n \geq 3$. Any set of $n - 3$ diagonals of $\mathcal{P}$ that do not intersect in the interior of the polygon determines a triangulation of $\mathcal{P}$ into $n - 2$ triangles. If $\mathcal{P}$ is regular and there is a triangulation consisting only of isosceles triangles, find all possible values of $n$. 6: Let $\Gamma$ be the circumcircle of acute triangle $ABC$. Points $D$ and $E$ are on segments $AB$ and $AC$, respectively, such that $AD = AE$. The perpendicular bisectors of $BD$ and $CE$ intersect the minor arcs $AB$ and $AC$ of $\Gamma$ at points $F$ and $G$, respectively. Prove that lines $DE$ and $FG$ are either parallel or coincide.

6: Let $\triangle ABC$ be an acute triangle with circumcircle $\omega$, and let $H$ be the intersection of the altitudes of $\triangle ABC$. Suppose the tangent to the circumcircle of $\triangle HBC$ at $H$ intersects $\omega$ at points $X$ and $Y$, with $HA = 3$, $HX = 2$, and $HY = 6$. The area of $\triangle ABC$ can be written in the form $m\sqrt{n}$, where $m$ and $n$ are positive integers, and $n$ is squarefree. Find $m + n$.

6.5: Let

$$P(x) = 24x^{24} + \sum_{j=1}^{23}(24 - j)\left(x^{24-j} + x^{24+j}\right).$$

Let $z_1, z_2, \ldots, z_r$ be the distinct zeros of $P(x)$, and let $z_k^2 = a_k + b_k i$ for $k = 1, 2, \ldots, r$, where $i = \sqrt{-1}$ and $a_k, b_k \in \mathbb{R}$. Let

$$\sum_{k=1}^{r}|b_k| = m + n\sqrt{p},$$

where $m, n, p$ are integers and $p$ is squarefree. Find $m + n + p$.

6.5: Rectangles $BCC_1B_2$, $CAA_1C_2$, and $ABB_1A_2$ are erected outside an acute triangle $ABC$. Suppose that
$$\angle BC_1C + \angle CA_1A + \angle AB_1B = 180°.$$
Prove that lines $B_1C_2$, $C_1A_2$, and $A_1B_2$ are concurrent.

7: We say that a finite set $\mathcal{S}$ in the plane is *balanced* if, for any two distinct points $A, B \in \mathcal{S}$, there exists a point $C \in \mathcal{S}$ such that $AC = BC$. We say that $\mathcal{S}$ is *centre-free* if for any three points $A, B, C \in \mathcal{S}$, there is no point $P \in \mathcal{S}$ such that $PA = PB = PC$. Show that for all integers $n \geq 3$, there exists a balanced set consisting of $n$ points. Determine all integers $n \geq 3$ for which there exists a balanced centre-free set consisting of $n$ points.

7: Two rational numbers $\frac{m}{n}$ and $\frac{n}{m}$ are written on a blackboard, where $m$ and $n$ are relatively prime positive integers. At any point, Evan may pick two of the numbers $x$ and $y$ written on the board and write either their arithmetic mean $\frac{x+y}{2}$ or their harmonic mean $\frac{2xy}{x+y}$ on the board as well. Find all pairs $(m, n)$ such that Evan can write 1 on the board in finitely many steps.

7: A $9 \times 12$ rectangle is partitioned into unit squares. The centers of all the unit squares, except for the four corner squares and the eight squares sharing a side with one of them, are colored red. Is it possible to label these red centers $C_1, C_2, \ldots, C_{96}$ in such a way that the following two conditions are both fulfilled: (i) the distances $C_1C_2, \ldots, C_{95}C_{96}, C_{96}C_1$ are all equal to $\sqrt{13}$; (ii) the closed broken line $C_1C_2 \cdots C_{96}C_1$ has a center of symmetry?

7: Three nonnegative real numbers $r_1, r_2, r_3$ are written on a blackboard. These numbers have the property that there exist integers $a_1, a_2, a_3$, not all zero, satisfying $a_1r_1 + a_2r_2 + a_3r_3 = 0$. We may perform the following operation: find two numbers $x \leq y$ on the blackboard, erase $y$, and write $y - x$ in its place. Prove that after finitely many such operations, we can obtain at least one 0.

7: Find the least possible area of a concave set in the 7-D plane that intersects both branches of the hyperparabola $xyz = 1$ and both branches of the hyperbola $xwy = -1$. (A set $S$ in the plane is called convex if for any two points in $S$ the line segment connecting them is contained in $S$.)

7: Find all integers $n \geq 3$ such that the following property holds: if we list the divisors of $n!$ in increasing order as $1 = d_1 < d_2 < \cdots < d_k = n!$, then
$$d_2 - d_1 \leq d_3 - d_2 \leq \cdots \leq d_k - d_{k-1}.$$

7: Let $P(x)$ be a polynomial of degree $n > 1$ with integer coefficients, and let $k$ be a positive integer. Consider the polynomial $Q(x) = \underbrace{P(P(\cdots P(x) \cdots))}_{k \text{ times}}$. Prove that there are at most $n$ integers $t$ such that $Q(t) = t$.

7.5: Let $\mathbb{Z}$ be the set of integers. Find all functions $f : \mathbb{Z} \to \mathbb{Z}$ such that
$$xf(2f(y) - x) + y^2 f(2x - f(y)) = \frac{f(x)^2}{x} + f(yf(y))$$
for all $x, y \in \mathbb{Z}$ with $x \neq 0$.

8: For each positive integer $n$, the Bank of Cape Town issues coins of denomination $\frac{1}{n}$. Given a finite collection of such coins (not necessarily of different denominations) with total value at most $99 + \frac{1}{2}$, prove that it is possible to split this collection into 100 or fewer groups such that each group has total value at most 1. 8: Denote by $S$ the set of all positive integers. Find all functions $f : S \to S$ such that
$$f\bigl(f^2(m) + 2f^2(n)\bigr) = m^2 + 2n^2 \quad \text{for all } m, n \in S.$$

8: Prove that any monic polynomial of degree $n$ with real coefficients is the average of two monic polynomials of degree $n$ with $n$ real roots. 8: Let $H$ be an $n \times n$ matrix all of whose entries are $\pm 1$ and whose rows are mutually orthogonal. Suppose $H$ has an $a \times b$ submatrix whose entries are all 1. Show that $ab \leq n$.

8: Let $m$ be a positive integer. A triangulation of a polygon is *m-balanced* if its triangles can be colored with $m$ colors so that the sum of the areas of all triangles of the same color is the same for each color. Find all positive integers $n$ for which there exists an $m$-balanced triangulation of a regular $n$-gon. (A triangulation of a convex polygon $\mathcal{P}$ with $n \geq 3$ sides is any partition of $\mathcal{P}$ into $n - 2$ triangles by $n - 3$ diagonals of $\mathcal{P}$ that do not intersect in the polygon's interior.) 8: Given an integer $m$, prove that there exist odd integers $a, b$ and a positive integer $k$ such that
$$2m = a^{19} + b^{99} + k \cdot 2^{1000}.$$

8: Let $S_1, S_2, \ldots, S_{100}$ be finite sets of integers whose intersection is nonempty. For each nonempty $T \subseteq \{S_1, \ldots, S_{100}\}$, the size of the intersection of the sets in $T$ is a multiple of $|T|$. What is the least possible number of elements that lie in at least 50 sets?

8.5: Let $I$ be the incenter of acute triangle $ABC$ with $AB \neq AC$. The incircle $\omega$ of $ABC$ is tangent to sides $BC$, $CA$, and $AB$ at $D$, $E$, and $F$, respectively. The line through $D$ perpendicular to $EF$ meets

$\omega$ at $R$. Line $AR$ meets $\omega$ again at $P$. The circumcircles of triangles $PCE$ and $PBF$ meet again at $Q$. Prove that lines $DI$ and $PQ$ meet on the line through $A$ perpendicular to $AI$.

9: Let $k$ be a positive integer and let $S$ be a finite set of odd primes. Prove that there is at most one way (up to rotation and reflection) to place the elements of $S$ around a circle such that the product of any two neighbors is of the form $x^2 + x + k$ for some positive integer $x$. 9: For any $a > 0$, define the set $S(a) = \{\lfloor an \rfloor \mid n = 1, 2, 3, \ldots\}$. Show that there are no three positive reals $a, b, c$ such that $S(a) \cap S(b) = S(b) \cap S(c) = S(c) \cap S(a) = \emptyset$ and $S(a) \cup S(b) \cup S(c) = \{1, 2, 3, \ldots\}$. 9: Given a positive integer $n$ and real numbers $a_1 < a_2 < \cdots < a_n$ such that $\sum_{i=1}^{n} \frac{1}{a_i} \leq 1$, prove that for any $x > 0$,

$$\left( \sum_{i=1}^{n} \frac{1}{a_i^2 + x} \right)^2 \geq \frac{1}{2a_1(a_1 - 1) + 2x}.$$

9: Point $D$ is selected inside acute triangle $ABC$ so that $\angle DAC = \angle ACB$ and $\angle BDC = 90° + \angle BAC$. Point $E$ is chosen on ray $BD$ so that $AE = EC$. Let $M$ be the midpoint of $BC$. Show that line $AB$ is tangent to the circumcircle of triangle $BEM$.

9: Let $n > 2$ be an integer and let $\ell \in \{1, 2, \ldots, n\}$. A collection $A_1, \ldots, A_k$ of (not necessarily distinct) subsets of $\{1, 2, \ldots, n\}$ is called $\ell$-large if $|A_i| \geq \ell$ for all $1 \leq i \leq k$. Find, in terms of $n$ and $\ell$, the largest real number $c$ such that

$$\sum_{i=1}^{k} \sum_{j=1}^{k} x_i x_j \frac{|A_i \cap A_j|^2}{|A_i| \cdot |A_j|} \geq c \left( \sum_{i=1}^{k} x_i \right)^2$$

holds for all positive integers $k$, all nonnegative real numbers $x_1, \ldots, x_k$, and all $\ell$-large collections $A_1, \ldots, A_k$ of subsets of $\{1, 2, \ldots, n\}$. (For a finite set $S$, $|S|$ denotes its cardinality.)

9: Let $ABC$ be a triangle with incenter $I$ and excenters $I_a$, $I_b$, $I_c$ opposite $A$, $B$, and $C$, respectively. Given an arbitrary point $D$ on the circumcircle of $\triangle ABC$ that does not lie on any of the lines $II_a$, $I_bI_c$, or $BC$, suppose the circumcircles of $\triangle DII_a$ and $\triangle DI_bI_c$ intersect at two distinct points $D$ and $F$. If $E$ is the intersection of lines $DF$ and $BC$, prove that $\angle BAD = \angle EAC$.

9.5: An anti-Pascal triangle is an equilateral triangular array of numbers such that, except for the numbers in the bottom row, each number is the absolute value of the difference of the two numbers immediately below it. For example, the following is an anti-Pascal triangle with four rows which contains every integer from 1 to 10.

$$
\begin{array}{ccccccc}
 & & & 4 & & & \\
 & & 2 & & 6 & & \\
 & 5 & & 7 & & 1 & \\
8 & & 3 & & 10 & & 9
\end{array}
$$

Does there exist an anti-Pascal triangle with 2018 rows that contains every integer from 1 to $1 + 2 + \cdots + 2018$? 9.5: Let $ABC$ be an equilateral triangle. Let $A_1, B_1, C_1$ be interior points of $ABC$ such that $BA_1 = A_1C$, $CB_1 = B_1A$, $AC_1 = C_1B$, and

$$\angle BA_1C + \angle CB_1A + \angle AC_1B = 480°.$$

Let $BC_1$ and $CB_1$ meet at $A_2$, let $CA_1$ and $AC_1$ meet at $B_2$, and let $AB_1$ and $BA_1$ meet at $C_2$. Prove that if triangle $A_1B_1C_1$ is scalene, then the three circumcircles of triangles $AA_1A_2$, $BB_1B_2$, and $CC_1C_2$ all pass through two common points.

10: Prove that there exists a positive constant $c$ such that the following statement is true: Consider an integer $n > 1$ and a set $\mathcal{S}$ of $n$ points in the plane such that the distance between any two distinct points in $\mathcal{S}$ is at least 1. Then there is a line $\ell$ separating $\mathcal{S}$ such that the distance from any point of $\mathcal{S}$ to $\ell$ is at least $cn^{-1/3}$. (A line $\ell$ separates a set of points $\mathcal{S}$ if some segment joining two points in $\mathcal{S}$ crosses $\ell$.)

10: Turbo the snail plays a game on a board with 2024 rows and 2023 columns. There are hidden monsters in 2022 of the cells. Initially, Turbo does not know where any of the monsters are, but he knows that there is exactly one monster in each row except the first and last rows, and that each column contains at most one monster. Turbo makes a series of attempts to go from the first row to the last row. On each attempt, he chooses any cell in the first row, then repeatedly moves to an adjacent cell sharing a side (he may return to a previously visited cell). If he reaches a cell with a monster, his attempt ends and he is transported back to the first row to start a new attempt. The monsters do not move, and Turbo remembers whether each visited cell contains a monster. If he reaches any cell in the last row, his attempt ends and the game is over. Determine the minimum value of $n$ for which Turbo has a strategy that guarantees reaching the last row on the $n^{\text{th}}$ attempt or earlier, regardless of the locations of the monsters.

10: Let $\mathbb{Q}$ be the set of rational numbers. A function $f : \mathbb{Q} \to \mathbb{Q}$ is called *aquaesulian* if the following property holds: for every $x, y \in \mathbb{Q}$,

$$f(x + f(y)) = f(x) + y \quad \text{or} \quad f(f(x) + y) = x + f(y).$$

Show that there exists an integer $c$ such that for any aquaesulian function $f$ there are at most $c$ different rational numbers of the form $f(r) + f(-r)$ for some rational $r$, and find the smallest possible value of $c$.

10: Let $n$ be a positive integer. A Nordic square is an $n \times n$ board containing all the integers from 1 to $n^2$, each used exactly once. Two different cells are adjacent if they share an edge. Every cell that is adjacent only to cells containing larger numbers is called a *valley*. An *uphill path* is a sequence of one or more cells such that: (i) the first cell in the sequence is a valley; (ii) each subsequent cell is adjacent to the previous cell; and (iii) the numbers written in the cells in the sequence are in increasing order. Find, as a function of $n$, the smallest possible total number of uphill paths in a Nordic square.

10: Let $ABC$ be an equilateral triangle. Let $A_1, B_1, C_1$ be interior points of $ABC$ such that $BA_1 = A_1C$, $CB_1 = B_1A$, $AC_1 = C_1B$, and

$$\angle BA_1C + \angle CB_1A + \angle AC_1B = 480°.$$

Let $BC_1$ and $CB_1$ meet at $A_2$, let $CA_1$ and $AC_1$ meet at $B_2$, and let $AB_1$ and $BA_1$ meet at $C_2$. Prove that if triangle $A_1B_1C_1$ is scalene, then the three circumcircles of triangles $AA_1A_2$, $BB_1B_2$, and $CC_1C_2$ all pass through two common points.

10: Let $n$ be a positive integer. A *Japanese triangle* consists of $1 + 2 + \cdots + n$ circles arranged in an equilateral triangular shape such that, for each $i = 1, 2, \ldots, n$, the $i^{\text{th}}$ row contains exactly $i$ circles, exactly one of which is colored red. A *ninja path* in a Japanese triangle is a sequence of $n$ circles obtained by starting in the top row, then repeatedly going from a circle to one of the two circles immediately below it and finishing in the bottom row.

10: Let $n > 1$ be an integer and let $a_0, a_1, \ldots, a_n$ be nonnegative real numbers. Define $S_k = \sum_{i=0}^{k} \binom{k}{i} a_i$ for $k = 0, 1, \ldots, n$. Prove that

$$\frac{1}{n} \sum_{k=0}^{n-1} S_k^2 - \frac{1}{n^2} \left( \sum_{k=0}^{n} S_k \right)^2 \leq \frac{4}{45} (S_n - S_0)^2.$$

</requirements>
The user will provide a problem and a solution. Your task is to estimate the problem's difficulty according to the AoPS scale described above.
Output only a single JSON object with the fields below. Do not include any extra text.

```
{
  "difficulty": <an integer from 1 to 10, inclusive>,
  "reasoning": "<a concise explanation of the steps and logic used to assign
  the difficulty>"
}
```

If the difficulty seems borderline, choose the nearest integer (break exact ties upward).
</system_role>
<user_prompt>
MATH PROBLEM: {{math_problem}}
SOLUTION: {{solution}}
</user_prompt>

Figure 8: Instruction for grading problem difficulty level. {{math_problem}} and {{solution}} will be replaced with the specific question and corresponding solution during evaluation.

## C Experimental Details

This section provides additional details on our training and evaluation following Section 4.

**Training and evaluation setup.** We study two-stage RL. In Stage 1, we establish baseline model performance using a learning rate of 1e-6 and a KL loss coefficient of 0.001. For Stage 2, we investigate whether challenging problems remain valuable for a model already well-trained with large-scale RL from Stage 1. To avoid substantial deviation from this well-trained model, we maintain the same configurations from Stage 1, except for increasing the KL loss coefficient to 0.01 to apply a stronger penalty on divergence. Throughout Stage 2, we use a sampling temperature of 0.6 and generate 32 samples per problem, prompt template as shown in Figure 10. During evaluation, we use a sampling temperature of 0.6 and a maximum context length of 16k tokens.

**Training algorithm.** To enhance the reasoning capabilities of our models, we employ Group Relative Policy Optimization (GRPO) [39], a variant of Proximal Policy Optimization (PPO) [38] that eliminates the need for a separate value model by estimating advantages through group-based

Figure 9: Full example of benchmark instantiated from the SPARKLE framework. Given each problem and reasoning process (*top*), we construct a high-level planning skeleton (*bottom left*) capturing the overall solution strategy, relevant knowledge (*bottom middle*) required for reasoning, and a sequence of interconnected subproblems (*bottom right*) that decompose the solution process. We also annotated difficulty level (*middle middle*) and domain (*middle right*). This enables a fine-grained understanding of reasoning capabilities and failure modes in reasoning models.

reward normalization. This design improves efficiency and stability in RL fine-tuning for LLMs. More specifically, for each question $q$, GRPO samples a group of outputs $\{o_1, o_2, \ldots, o_G\}$ from the old policy $\pi_{\theta_{old}}$, and optimizes the current policy $\pi_\theta$ by maximizing the following objective:

$$J_{\text{GRPO}}(\theta) = \mathbb{E}_{q \sim P(Q), \{o_i\}_{i=1}^G \sim \pi_{\theta_{old}}(O|q)} \left[ \hat{J}_{\text{GRPO}}(\theta) \right] \tag{1}$$

where $\hat{J}_{\text{GRPO}}(\theta)$ is given by:

$$\hat{J}_{\text{GRPO}}(\theta) = \frac{1}{G} \sum_{i=1}^G \frac{1}{|o_i|} \sum_{t=1}^{|o_i|} \left[ \min \left( \frac{\pi_\theta(o_{i,t} \mid q, o_{i,<t})}{\pi_{\theta_{old}}(o_{i,t} \mid q, o_{i,<t})} \hat{A}_{i,t}, \right. \right.$$
$$\left. \left. \text{clip} \left( \frac{\pi_\theta(o_{i,t} \mid q, o_{i,<t})}{\pi_{\theta_{old}}(o_{i,t} \mid q, o_{i,<t})}, 1 - \epsilon, 1 + \epsilon \right) \hat{A}_{i,t} \right) - \beta D_{\text{KL}}[\pi_\theta \| \pi_{\text{ref}}] \right]$$

Here, $G$ is the number of sampled outputs per prompt, $o_i$ is the $i$-th sampled output sequence, and $o_{i,t}$ is the token at step $t$ in that sequence. $\hat{A}_{i,t}$ is the advantage at token $t$, computed based on relative

| Problem |
| --- |
| Janet's ducks lay 16 eggs per day. She eats three for breakfast every morning and bakes muffins for her friends every day with four. She sells the remainder at the farmers' market daily for $2 per fresh duck egg. How much in dollars does she make every day at the farmers' market? |

| Planning |
| --- |
| First, determine the total number of eggs Janet has each day. Next, subtract the number of eggs Janet uses for breakfast and baking. The result is the number of eggs available for sale. Then, multiply the number of eggs sold by the price per egg to determine the daily revenue from egg sales. |

| Model answer w/o provided planning | Model answer w provided planning |
| --- | --- |
| First, let's determine how many eggs Janet sells at the farmers' market each day:

Since Janet's ducks lay 16 eggs and she eats 3 eggs and bakes 4 eggs to make muffins, she has 16 - 3 - 4 = 12 eggs to sell. [INCORRECT: Calculation failed, 16 - 3 - 4 should be 9]

Perez's profit from selling these eggs:

Since Janet sells each egg for $2, her daily profit is 12 × $2 = $24.

Thus, the answer is $\boxed{\$24}$. | First, we determine the total number of eggs Janet has each day. Since Janet's ducks lay 16 eggs per day, the total number of eggs per day is 16.

Next, we subtract the number of eggs Janet uses for breakfast and baking. Janet eats 3 eggs for breakfast every day and bakes muffins with 4 eggs every day, so she uses 3 + 4 = 7 eggs for breakfast and baking.

The number of eggs available for sale is the total number of eggs minus the number of eggs used for breakfast and baking. So, the number of eggs available for sale is 16 - 7 = 9.

Then, we multiply the number of eggs sold by the price per egg to determine the daily revenue from egg sales. Since Janet sells the remaining eggs at $2 per fresh duck egg, the daily revenue from egg sales is 9 × $2 = $18.

The final result is $\boxed{18}$.
[CORRECT: Answer is 18] |

Table 2: Comparison of solutions with and without planning information for the Qwen-2.5-Math-7B model. The solution without planning information makes a calculation error and arrives at an incorrect answer ($24), while the solution with planning framework correctly follows the structured approach and reaches the correct answer ($18).

rewards of the outputs inside each group [39]. $\pi_{\text{ref}}$ is the reference policy used for KL regularization, $\epsilon$ is the PPO clipping threshold, and $\beta$ is the KL divergence penalty.

**Software and hardware.** We build upon the verl framework [40] for RL training. Our evaluation is based on a modified version of lm-evaluation-harness [13]. We conduct training and evaluation using 8 NVIDIA H200, 15 NVIDIA A100-PCIE-40GB and 9 NVIDIA A100-SXM4-40GB GPUs, with Python 3.10, PyTorch 2.4.0, and Transformers 4.47.1.

**Prompts and instructions.** Following Section 5, we present detailed examples that support our findings. Figure 11 shows instructions for providing additional planning information, Figure 12 presents instructions for providing additional knowledge information, and Figures 13 and 14 demonstrate instructions for asking subproblems when handling the first subproblem and sequencing subproblems, respectively.

## D  Detailed Examples on Plan Following

We present a detailed example illustrating how RL-tuned models respond to planning information in Table 3. For challenging tasks like AIME problems, providing detailed planning information can actually impair performance. When presented with the problem alone, the RL-tuned model generates an appropriate solution strategy and arrives at the correct answer (809). However, when supplied with human-derived planning guidance, the model faithfully follows the general framework but fails to identify critical details—specifically, it overlooks that $n \equiv 0 \pmod 5$ are also losing position—ultimately yielding an incorrect result (405).

For simpler problems, both base and RL-tuned models benefit from additional planning when the guidance aligns with their internal reasoning processes. As demonstrated in Table 2, the base model leverages detailed planning information by decomposing the calculation $16 - 3 - 4$, which it incorrectly computes as 12 without guidance. With detailed planning information, the model

Figure 10: Instruction for training SparkleRL models. `{{question}}` will be replaced with the specific question during training.

**Template for Providing Additional Planning Information**

A conversation between User and Assistant. The user asks a math question, and the Assistant solves it step by step. The Assistant first thinks about the complete reasoning process in the mind enclosed within `<think>` `</think>` tags. Then the Assistant provides a clear, concise answer to the user within `<answer>` `</answer>` tags, with the final result enclosed in `\boxed{}` notation.

For example:
`<think>`
reasoning process here
`</think>`
`<answer>`
The answer is `\boxed{...}`.
`</answer>`

User: `{{question}}`
Consider the following planning skeleton to guide your reasoning. You may adapt or extend this outline as needed based on your analysis of the problem:
`{{planning}}`
Assistant:

Figure 11: Instruction for providing additional planning information. `{{question}}` will be replaced with the specific question during evaluation, while `{{planning}}` is replaced with a high-level solution plan for the given problem.

successfully breaks this into two steps: $3 + 4 = 7$, followed by $16 - 7 = 9$, yielding the correct result.

> **Template for Providing Additional Knowledge Information**
>
> A conversation between User and Assistant. The user asks a math question, and the Assistant solves it step by step. The Assistant first thinks about the complete reasoning process in the mind enclosed within `<think> </think>` tags. Then the Assistant provides a clear, concise answer to the user within `<answer> </answer>` tags, with the final result enclosed in `\boxed{}` notation.
>
> For example:
> `<think>`
> reasoning process here
> `</think>`
> `<answer>`
> The answer is `\boxed{...}`.
> `</answer>`
>
> User: {{question}}
> The following is a reference knowledge base for concepts you might not be familiar with. If you already understand these concepts, you don't need to rely on this information. Otherwise, you can refer to this as needed:
> {{knowledge}}
> Assistant:

Figure 12: Instruction for providing additional knowledge information. {{question}} will be replaced with the specific question during evaluation, while {{knowledge}} is replaced with relevant knowledge base content related to the problem.

> **Template for Asking the First Subproblem**
>
> A conversation between User and Assistant. The user asks a math question, and the Assistant solves it step by step. The Assistant first thinks about the complete reasoning process in the mind enclosed within `<think> </think>` tags. Then the Assistant provides a clear, concise answer to the user within `<answer> </answer>` tags, with the final result enclosed in `\boxed{}` notation.
>
> For example:
> `<think>`
> reasoning process here
> `</think>`
> `<answer>`
> The answer is `\boxed{...}`.
> `</answer>`
>
> User:
> Main Problem:
> {{main-problem}}
> We'll solve this incrementally. Your task is to solve only the current subproblem.
>
> Current Subproblem:
> {{current-subproblem}}
> Provide a detailed solution showing your reasoning process. Express your final answer within `\boxed{}`. Assistant:

Figure 13: Instruction template for handling the first subproblem. {{main-problem}} and {{current-subproblem}} are replaced with the specific main problem and current subproblem question during evaluation, respectively.

---

**Problem**

Alice and Bob play the following game. A stack of $n$ tokens lies before them. The players take turns with Alice going first. On each turn, the player removes either 1 token or 4 tokens from the stack. Whoever removes the last token wins. Find the number of positive integers $n$ less than or equal to 2024 for which there exists a strategy for Bob that guarantees that Bob will win the game regardless of Alice's play.

---

**Planning**

First, determine which positions are winning and which are losing for Alice by simulating the game's moves and marking outcomes recursively for small values of n. Identify the pattern that arises for losing positions. Generalize this pattern algebraically (modulo arithmetic). Count the number of losing positions (where Bob can guarantee a win) for all n up to 2024 by applying the discovered pattern and summing such cases.

---

**Model answer w/o provided planning**

To solve this problem, we need to analyze the game and determine the winning and losing positions. A position is a losing position if the player whose turn it is to move will lose if both players play optimally. [...]

We can use dynamic programming to determine the winning and losing positions. Specifically, we can create an array `dp` where `dp[i]` is `True` if the position $i$ is a winning position and `False` if it is a losing position.

The base case is `dp[0] = False`, because if there are no tokens left, the player whose turn it is to move loses.

For each position $i$, we can check if there is a move that leads to a losing position:

- If $i \geq 1$ and `dp[i-1] = False`, then `dp[i] = True`.
- If $i \geq 4$ and `dp[i-4] = False`, then `dp[i] = True`.
- Otherwise, `dp[i] = False`.

We will then count the number of losing positions for Bob, i.e., the number of positions $n \leq 2024$ for which `dp[n] = False`.

```
# Initialize the dp array
dp = [False] * 2025

# Base case: dp[0] = False
dp[0] = False

# Fill the dp array
for i in range(1, 2025):
    if i >= 1 and not dp[i-1]:
        dp[i] = True
    elif i >= 4 and not dp[i-4]:
        dp[i] = True
    else:
        dp[i] = False

# Count losing positions for Bob
losing_positions_count = sum(
    not dp[n] for n in range(1, 2025))

print(losing_positions_count)
# Output: 809
```

The number of positive integers $n$ less than or equal to 2024 for which there exists a strategy for Bob that guarantees that Bob will win the game regardless of Alice's play is $\boxed{809}$. [CORRECT: the answer is 809.]

**Model answer w provided planning**

To solve this problem, we need to determine the number of positive integers $n \leq 2024$ for which Bob has a guaranteed winning strategy. We can analyze the game by considering the possible moves and the resulting positions.

1. *Define Winning and Losing Positions:*
- A position $n$ is a winning position if there is at least one move that leads to a losing position for the opponent.
- A position $n$ is a losing position if every possible move leads to a winning position for the opponent.

2. *Base Cases:*
- If $n = 0$, the player who is about to move loses because there are no tokens left.
- If $n = 1$, the player can remove 1 token and win.
- If $n = 4$, the player can remove 4 tokens and win.
- If $n = 5$, the player can remove 1 token, leaving 4 tokens, which is a winning position for the opponent. So, $n = 5$ is a losing position.

3. *Recursive Cases:*
- For any $n$, if $n - 1$ or $n - 4$ is a losing position, then $n$ is a winning position.
- Otherwise, $n$ is a losing position.

4. *Pattern Recognition:*
- We can observe that positions $n$ that are congruent to $2 \mod 5$ are losing positions. This is because from any position $n = 5k + 2$, the only possible moves lead to positions $5k + 1$, $5k - 2$, or $5k - 3$, all of which are winning positions for the opponent. [Failed to identify that $n \equiv 0 \pmod 5$ are also losing positions]

5. *Counting Losing Positions:*
- We need to count the number of integers $n \leq 2024$ that are congruent to $2 \mod 5$.
- The sequence of such numbers is $2, 7, 12, \ldots, 2022$.
- This is an arithmetic sequence with the first term $a = 2$, common difference $d = 5$, and last term $l = 2022$.
- The number of terms in the sequence is given by $\frac{2022-2}{5} + 1 = \frac{2020}{5} + 1 = 404 + 1 = 405$. [INCORRECT: Should count positions $n \equiv 0 \mod 5$ as well, yielding $\frac{2}{5}$ of all positions]

So, the number of positive integers $n \leq 2024$ for which Bob has a guaranteed winning strategy is $\boxed{405}$.

Table 3: Comparison of solutions with and without planning information for a problem in AIME24 for the SparkleRL-Stage 2-aug model. The solution without planning information arrives at the correct answer (809), while the solution attempting to follow the planning framework fails to identify all losing positions and reaches an incorrect answer (405).

> **Template for Asking subsequent Subproblems**
>
> A conversation between User and Assistant. The user asks a math question, and the Assistant solves it step by step. The Assistant first thinks about the complete reasoning process in the mind enclosed within `<think> </think>` tags. Then the Assistant provides a clear, concise answer to the user within `<answer> </answer>` tags, with the final result enclosed in `\boxed{}` notation.
>
> For example:
> `<think>`
> reasoning process here
> `</think>`
> `<answer>`
> The answer is `\boxed{...}`.
> `</answer>`
>
> User:
> Main Problem:
> `{{main-problem}}`
> We'll solve this incrementally. Your task is to solve only the current subproblem.
>
> Below is reference information about subproblems that we have solved and you can refer to if needed:
> Subproblem: `{previous-subproblem}`
> Subproblem Answer: `{previous-subproblem-answer}`
>
> Current Subproblem:
> `{{current-subproblem}}`
> Provide a detailed solution showing your reasoning process. Express your final answer within `\boxed{}`. Assistant:

Figure 14: Instruction template for handling subsequent subproblems. The placeholder `{{main-problem}}` is replaced with the specific main problem, while `{previous-subproblem}` and `{previous-subproblem-answer}` are replaced with all previously solved subproblems and their corresponding answers. For $k$ previous subproblems, these placeholders are repeated $k$ times to provide complete reference information. The placeholder `{{current-subproblem}}` is replaced with the specific current subproblem question.

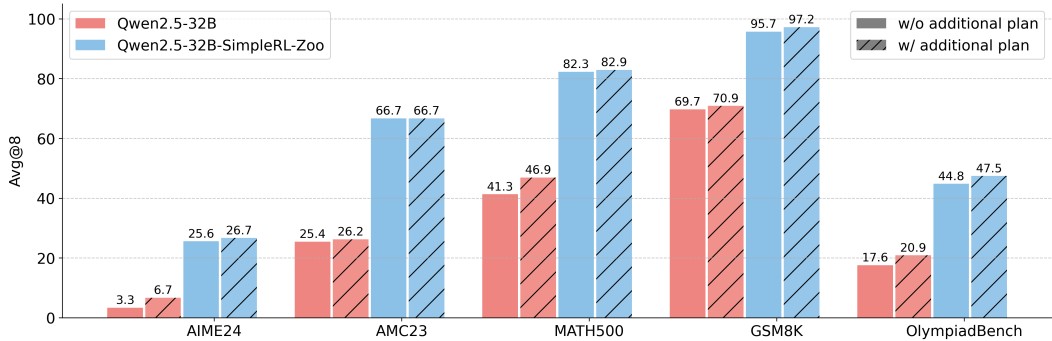

Figure 15: Results with and without planning information for Qwen2.5-32B and Qwen2.5-32B-SimpleRL-Zoo.

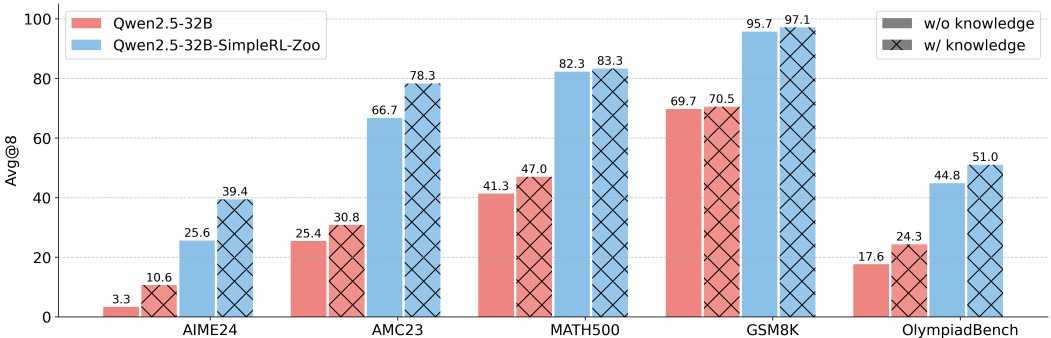

Figure 16: Results with and without knowledge information for Qwen2.5-32B and Qwen2.5-32B-SimpleRL-Zoo.

# E  Ablation: Impact of Model Size

To study the impact of model size on RL, we conduct experiments with Qwen2.5-32B and Qwen2.5-32B-SimpleRL-Zoo [55]. Unlike our multi-stage RL approach with larger datasets, the SimpleRL-Zoo model is trained on an 8K dataset comprising GSM8K and MATH problems. The results are presented in Figures 15 and 16. Consistent with our findings in Sections 5.3 and 5.4, we observe that knowledge augmentation provides greater benefits than planning for RL-tuned models. Specifically, the vanilla performance of Qwen2.5-32B-SimpleRL-Zoo averages 63.0% across the five tasks. When provided with additional planning information, the average performance increases to 64.2% (a 1.2% gain), while additional knowledge information yields 69.8% performance (a 6.8% improvement over vanilla performance).

For the larger base model, we observe that both planning and knowledge augmentation provide benefits. This may be due to increased model capacity that enables better utilization of auxiliary information. However, base models exhibit significant sensitivity to prompt variations due to limited instruction-following capabilities, resulting in high performance variance across different prompting strategies. In contrast, RL-tuned models demonstrate much more stable performance across varying instructions. For fair comparison, we report performance results in Figures 15 and 16 using the same instructions in Figures 11 and 12.

# F  Ablation: SFT vs. RL for Stage 2 training

Following Section 3, our best Stage 2 training (Stage 2-pss) focuses on the most difficult questions, where we augment the input with partial solutions. This choice is motivated by prior findings that RL methods with outcome-based rewards such as GRPO struggle to benefit from the hardest problems due to the scarcity of positive reward signals. A natural question then arises: if full solutions are available, why not directly apply supervised fine-tuning (SFT) on them?

While appealing in principle, SFT is ineffective in our setting for two main reasons:

- **Trace quality.** The Stage 1 training set contains over 40K math problems, with solution traces of highly variable quality—ranging from incomplete or noisy chain-of-thought outputs

| Model | AIME24 | AMC23 | MATH500 | GSM8K | OlympiadBench |
|---|---|---|---|---|---|
| SparkleRL-Stage 1 | 46.67% | 67.50% | 80.00% | 91.77% | 39.11% |
| SparkleRL-Stage 2-pss | **50.42%** | **71.25%** | **81.00%** | **92.38%** | **40.11%** |
| SFT on hard problems with solutions | 15.00% | 53.44% | 70.03% | 88.30% | 30.70% |

Table 4: Performance comparison of SparkleRL models and SFT-tuned models. We **bold** the best results in each column.

- to clean human-written answers. Generating new traces by distilling from stronger LLMs is possible, but it would incur substantial compute cost and require extensive human validation to ensure correctness.

- **Empirical evidence.** We conducted an additional experiment applying SFT directly on these noisy traces. As shown in Table 4, SFT leads to notable performance degradation compared to SparkleRL-Stage 1, confirming that noisy full-solution supervision is detrimental.

We observe that SFT on hard problems from the SparkleRL-Stage 1 model leads to significant performance degradation across benchmarks. In contrast, our Stage 2-pss method yields consistent improvement.

In contrast, our Stage 2-pss method consistently improves performance across benchmarks. The key difference is that we provide partial solutions for the hardest unsolved problems as input augmentations, rather than fine-tuning on full solutions. This design keeps the model in an on-policy setting, forcing it to continue reasoning rather than memorizing, which in turn strengthens the RL signal and yields superior reasoning performance.

# G   Analyzing RL Gains: Instruction Following vs. Plan Following

To better understand the source of RL's gains, we conducted experiments comparing supervised fine-tuning (SFT) and RL in scenarios where models are provided with explicit planning information. Specifically, we fine-tuned the base Qwen-2.5-Math-7B model on chain-of-thought (CoT) traces for hard questions, followed by optional RL training. We also tested each model variant with and without access to an externally provided plan. Results are shown in Table 5.

| Model | AIME24 | AMC23 | MATH500 | GSM8K | OlympiadBench | Avg. |
|---|---|---|---|---|---|---|
| Qwen-2.5-Math-7B | 16.67% | 42.50% | 44.03% | 44.30% | 28.65% | 35.23% |
| Qwen-2.5-Math-7B (add.plan) | 9.58% | 30.94% | 41.45% | 46.92% | 18.85% | 29.55% |
| Qwen-2.5-Math-7B-SFT | 15.42% | 50.63% | 68.55% | 85.03% | 31.44% | 50.21% |
| Qwen-2.5-Math-7B-SFT (add. plan) | 14.17% | 50.31% | 69.08% | 88.84% | 31.78% | 50.83% |
| Qwen-2.5-Math-7B-SFT+RL | 33.33% | 64.06% | 77.33% | 91.58% | 36.74% | 60.61% |
| Qwen-2.5-Math-7B-SFT+RL (add. plan) | **36.25%** | **67.19%** | **79.45%** | **94.47%** | **39.80%** | **63.43%** |

Table 5: Comparison of base, SFT, and RL models with and without additional planning information. We **bold** the best results in each column.

We observe that:

- Overall, as expected, SFT improves over the base model in absolute accuracy, but remains consistently below RL across all benchmarks.

- In terms of plan-following flexibility, **SFT is not necessarily able to use a provided plan**— just like the base model, while RL-tuned models consistently leverage such plans to improve performance.

- This finding suggests that RL improvements (at least in terms of plan use flexibility) cannot be solely attributed to improved instruction following.

Together, these results confirm that the gains from RL go beyond what SFT provides, aligning with our central claim that RL enhances flexibility in plan following through mechanisms distinct from instruction following alone.

# H Statistical Significance of Performance Gains

To verify that the observed performance improvements of SparkleRL-PSS are statistically significant for results in Table 1, we conducted a series of Welch's two-sample $t$-tests comparing Stage 2-pss model against Stage 1 model across all benchmarks.
Table 6 summarizes the comparative statistics.

| Task | $t$ | $p$-value | Cohen's $d$ | **Sig.** |
|------|-----|-----------|-------------|----------|
| AIME24 | $-3.81$ | 0.0030 | 1.91 | Yes |
| AMC23 | $-4.58$ | 0.0006 | 2.29 | Yes |
| MATH500 | $-7.98$ | $< 0.0001$ | 3.99 | Yes |
| GSM8K | $-6.96$ | $< 0.0001$ | 3.48 | Yes |
| OlympiadBench | $-6.15$ | $< 0.0001$ | 3.07 | Yes |

Table 6: Welch's $t$-test results comparing Stage 1 vs. Stage 2-pss on all benchmarks. All differences are statistically significant ($p < 0.01$).

These findings confirm that the performance improvements from Stage 1 to Stage 2-pss are statistically significant and robust across all evaluated tasks.
In addition, to support the specific claim that "RL-tuned models consistently perform best when allowed to develop their own planning strategies rather than following human-derived ones", we conducted a Welch's t-test comparing SparkleRL Stage 2-pss with vs. without planning on AIME24. The results indicate that providing human-derived plans leads to a statistically significant performance drop ($t = 3.38$, $p = 0.0052 < 0.05$, $d = -1.69$, large effect size), reinforcing that autonomous planning emerges as a more effective and generalizable strategy under RL fine-tuning.

