# OpenReview forum: "Beyond Accuracy: Dissecting Mathematical Reasoning for LLMs Under Reinforcement Learning"
_NeurIPS.cc/2025/Conference — NeurIPS 2025 poster_

### Official Review · Reviewer_qeem · 2025-06-09

**Clarity:** 2
**Significance:** 2
**Originality:** 2
**Rating:** 3
**Confidence:** 5

**Summary:**

This paper employs LLMs to synthesize three complementary components for mathematical questions, namely, the planning skeleton, relevant knowledge, and sub - problems. By leveraging these complementary elements, along with the original questions, as input information, the paper attempts to analysis diverse aspects of the performance of LLMs in mathematical reasoning, extending beyond mere accuracy.
This paper adopts a multi-stage training approach to explore the influence of question difficulty on the efficacy of training. Through the augmentation of partial-solution guidance, challenging problems can be made relevant for reinforcement learning (RL) training.

**Questions:**

1. How do the multi-staged training experiments integrate within your Sparkle framework? They appear to involve distinct objectives being presented within a single research paper. It seems as if two experiments are combined to amass the contributions of this paper.
2. Given that, when complementary information is used as input, a decline in performance is observed in SFT models. Does this not suggest that the instructions are of poor quality? The complementary information is generated by LLMs, which may contain toxic and distracting elements. Does the absence of human verification not undermine the effectiveness of your framework?
3. The analysis framework is contingent upon the quality of the data synthesized by this LLM. This data serves as the gold standard for each aspect of reasoning, which can be brittle and biased. How do you ensure fairness and prevent bias?
4. How do you justify the conduction of providing partial solutions to the RL process? Partial solutions may train models to rely on step-by-step hints, reducing autonomy—contradicting RL’s goal of fostering self-generated strategies. Why not SFT instead if you have the "gold standard" solution? Could SFT models perform better with human-authored (non-LLM) plans? The paper assumes LLM-generated plans are "valid," but human planners might structure steps differently.

**Ethical Concerns:**

["NO or VERY MINOR ethics concerns only"]

**Limitations:**

data dependency assumptions (LLM-generated gold standards) and under-explored tradeoffs in training design (e.g., partial solutions)

**Quality:**

2

**Strengths And Weaknesses:**

Strenghts:
The paper is well written.
Analysis of mathmatical reasoning beyond accuracy is a welcoming topic

Weaknesses:
Overreliance on LLM-generated annotations for metrics analysis, relying on a single LLM (GPT-4.1) for annotation may embed biases and artifacts.
The claim that "RL improves plan flexibility" is conflated with "SFT struggles with LLM-generated plans," leaving open whether the deficit is model- or input-specific.
No direct comparison between LLM-generated vs. human-generated plans leaves ambiguity on input quality’s role.

---

> ### Author Rebuttal · Authors · 2025-07-31
>
> We appreciate your comments and suggestions and we address each question and provide further clarifications in detail below. Please let us know if you have any additional questions or comments and we are more than happy to discuss further.
>
> > Q1: How do the multi-staged training experiments integrate within Sparkle framework? They appear to involve distinct objectives being presented within a single research paper.
>
> Thank you for the thoughtful question. The dual focus is intentional and essential to our study. Our work aims to understand mathematical reasoning under RL. When applying RL to a set of problems, the key knobs are (i) the components necessary to solve a problem (planning, knowledge, etc.)—captured through the SPARKLE analysis framework—and (ii) the raw difficulty of the problems, which motivates our use of multi-stage RL training. A full characterization of RL for reasoning is not possible without addressing both of these elements.
>
> To unify these aspects, we adopt a curriculum-style two-stage RL training setup, inspired by prior works (Sec. 2), and analyze its effects of RL using the SPARKLE framework.  Specifically, Stage 1 establishes a strong RL-tuned model using diverse math problems, while Stage 2 explores three strategies for fine-tuning on a subset of the most challenging problems. We explore three data strategies (L184-188) for the second-stage training, from naive RL training with no augmentation, to including partial-solution augmentation. Our results show that without augmentation, outcome-based RL struggles to learn from difficult problems due to sparse or absent reward signals (Table 1). In contrast, partial-solution augmentation emerges as a simple yet effective method for enabling RL to extract training signal from these challenging cases.
>
> The SPARKLE framework is applied consistently across models trained at each stage, allowing us to systematically analyze how different training strategies influence specific aspects of reasoning. Rather than representing distinct objectives, the RL experiments and SPARKLE framework are tightly coupled to provide a unified and comprehensive analysis of RL’s impact on LLM reasoning.
>
> > Q2.1: Given that, when complementary information is used as input, a decline in performance is observed in SFT models. Does this not suggest that the instructions are of poor quality?
>
> We would like to clarify that **our paper does not evaluate or include SFT models**. To avoid confounding effects from supervised fine-tuning, we focus exclusively on *comparing base models with RL-tuned models*. This design complements prior studies that analyze the interplay between SFT and RL [1,2] (l.194–195).
>
> All results involving planning and knowledge interventions (e.g., Sec. 6.3 and 6.4) are based on this Base vs. RL-tuned comparison. (summarized in l.220–237).
>
> In Sec. 6.3, we find that providing external plans can degrade performance—particularly on the most challenging benchmark (AIME24). However, this is **not** due to poor instruction quality, but rather reflects a **misalignment** between externally provided plans and the model's preferred internal plans.
>
> Moreover, the base model itself exhibits some instruction-following ability: for example, its performance improves with planning on GSM8K (Fig. 3, l.279–284), and with sufficient sampling (Fig. 2), it can match RL-tuned model performance.
>
>
> > Q2.2&Q3: The complementary information is generated by LLMs, which may contain toxic and distracting elements...Does the absence of human verification not undermine the effectiveness of your framework?
>
> **Human verification** is an integral part in the benchmark construction process and has been included in this work.
>
> We would like to clarify that we utilize a web agent to "extract a planning skeleton and summarizing key reasoning steps" (l.145-146) from **existing step-by-step solutions in the test sets**, instead of prompting LLM to generate a plan *without reference*. Moreover, we address this risk through a rigorous multi-stage annotation and verification pipeline, as described in Sec. 3.2 (l.149–153). Specifically:
> - **A second verification agent** checks each generated annotation for correctness, coherence, completeness, and pedagogical soundness. If any aspect fails, the annotation is regenerated.
>
> - **Final validation is conducted by graduate students** with advanced mathematics backgrounds to ensure that each annotation faithfully captures the required reasoning and knowledge.
>
>
> In addition, to curate the training set for the second-stage RL tuning, we also involved "human verification to further filter out items with flawed solutions" (l.207-208). We believe this step is important for mitigating the potential brittleness of the GPT-4.1-based web agent.
>
>
> We hope this clarifies that human verification is not only present but plays a critical role in ensuring the reliability and effectiveness of our framework. We would appreciate if you could re-evaluate our paper in light of the human verification measures we have in place.
>
> > Q4: How do you justify the inclusion of partial solutions in RL? Partial solutions may train models to rely on step-by-step hints, reducing autonomy—contradicting RL’s goal of fostering self-generated strategies.
>
> Thank you for raising this point. Our motivation is grounded in prior work showing that curriculum-based approaches can significantly enhance the effectiveness of RL training for LLMs (see Sec. 2). However, outcome-based RL methods such as GRPO often struggle to learn from the most challenging problems due to the sparsity of positive reward signals (i.e., most sampled trajectories in a group result in incorrect answers).
>
> This raises the question: can we still leverage these hard problems in a way that improves performance *without compromising the capabilities already learned by a well-tuned RL model*?
>
>
> We agree that overusing partial solutions may reduce autonomy if applied indiscriminately. However, our approach is **targeted and carefully scoped**. Partial-solution augmentation is **not** used as a general strategy for RL training (i.e., Stage 1), which could encourage dependency on hints. Instead, we introduce partial-solution scaffolding **only in Stage 2**, and **only on the hardest problems** that the Stage 1 RL-tuned model consistently fails to solve.
>
> Therefore, We would like to clarify that partial solution augmentation is not used as a general approach for RL training on all problems (Stage 1), which "may train models to rely on step-by-step hints". We only perform partial solution scaffolding in the second stage, **on the hardest problems the first-stage RL-tuned model cannot solve**, to further boost the reasoning performance.
>
> Specifically, given an RL-trained model, we have explored three data strategies for the second-stage RL training (l.184-188):
>
> - (1) Hard-Only: Training on difficult problems without any augmentation.
>  - (2) Mixed Difficulty: Combining easy and hard problems for exposure diversity.
>  - (3) Hard-Augmented: Introducing partial solutions to help the model navigate complex reasoning paths.
>
> Our results (Table 1) show that the Hard-Augmented strategy significantly outperforms the other two, demonstrating that partial-solution scaffolding can effectively improve reasoning when the model is **otherwise unable to learn from sparse feedback**. In this context, partial augmentation serves as a practical curriculum tool to extend the learning frontier—rather than undermining reasoning autonomy.
>
>
>
> > Q5: Why not SFT instead if you have the "gold standard" solution?
>
> We agree that a natural baseline would be to apply SFT using available solution traces. However, there are both **practical limitations** and **empirical drawbacks** that make SFT sub-optimal in our setting.
>
> - The training set used in Stage 1 contains over 40K math problems, for which "gold standard" solutions are largely unreliable: many solution traces are noisy or incomplete. These traces vary in quality---from simplistic chain-of-thought outputs to human-written final answers. While obtaining new traces by distilling from competitive LLMs is possible, it would require substantial compute and additional human validation to ensure correctness and clarity.
> - Empirically, we observe that applying SFT on noisy traces does not lead to improved performance. To evaluate this, we conducted a controlled comparison between three approaches:
>
>
> | Model                     | AIME24     | AMC23      | MATH500    | GSM8K      | OlympiadBench |
> | ------------------------- | ---------- | ---------- | ---------- | ---------- | ------------- |
> |**SparkleRL-Stage 1**| 46.67%| 67.50%| 80.00%     |91.77%|39.11%|
> |**SparkleRL-Stage 2-aug** | **50.42%**| **71.25%**| **81.00%**|**92.38%**| **40.11%**|
> |**SFT on hard problems with solutions**|15.00%|53.44%|70.03%|88.30%|30.70%|
>
>
>
> As shown above, SFT on hard problems from the SparkleRL-Stage 1 model leads to significant **performance degradation** across all benchmarks. In contrast, our **Stage 2-aug** method yields consistent improvement.
>
> The key distinction in Stage 2-aug (details in l.202–216) is that instead of fine-tuning on full traces that are noisy, we augment the hardest unsolved examples from Stage 1 with partial solutions in the input prompt. These partial solutions does not disrupt agent's on-policy learning, which is important for effective RL training. This preserves the benefits of RL, which encourages models to develop robust, self-directed reasoning strategies—given hints (partial solutions) from the challenging problems.
>
> In contrast, SFT fine-tuning on noisy or overly prescriptive solutions can misalign with the model's learned heuristics, disrupting the internal reasoning behaviors acquired during RL training.
>
>
>
> [1] Chu et al., Sft memorizes, rl generalizes: A comparative study of foundation model post-training, 2025
>
> [2] Yeo et al., Demystifying long chain-of-thought reasoning in llms, 2025.

---

### Official Review · Reviewer_1rRb · 2025-06-30

**Clarity:** 3
**Significance:** 3
**Originality:** 3
**Rating:** 5
**Confidence:** 4

**Summary:**

This paper analyzes various interventions in RLVF posttraining on top of Qwen-2.5-Math-7B in order to better understand what limits reasoning performance: planning, knowledge, problem decomposition, or problem difficulty. The paper proposes a new dataset of 2,564 questions called SPARKLE which augments popular math benchmarks (such as GSM8K, MATH, AIME, etc.) with (1) a suggested plan for solving the problem; (2) relevant knowledge/concepts needed to solve the problem;  and (3) a set of subproblems and corresponding answers that need to be solved in the course of solving the whole problem. Ablations that either include or exclude the plans, knowledge, and subproblems demonstrate that: plans harm the base model’s performance while leaving the RL-trained model performance unchanged; knowledge harms base model performance but slightly improves RL-trained model performance; and all models seem unable to reliably solve subproblems. Additionally, the paper demonstrates that continued RL training on the most difficult problems augmented with hints can lead to the best performance.

**Questions:**

### Unsupported or overly broad claims

* I think the drops in performance seen by the base model might just be due to poor instruction following. I therefore think that (1) the paper should state this is the reason for the drop in base model performance, and (2) Qwen-2.5-Math-7B-Instruct should really be included as a baseline in order to disentangle whether it is the lack of instruction-tuning that leads to these drops or the lack of reasoning. My prediction is actually that including things like the plan will help Instruct quite a lot, but maybe not as much as doing RLVF (i.e. base \< instruct \< instruct+plan \< rl $\\approx$ rl+plan). **I would need to see Instruct added as a baseline to consider increasing my score**.
* Some of the claims need to be backed up by statistical tests—e.g. In the discussion in paragraph starting line 285, it’s not clear to me that the increases/decreases in performance are statistically significant. If you would like to make claims like “RL-tuned models achieve superior performance when permitted to develop autonomous planning strategies” then it is important to establish whether that superior performance is statistically significant. (Also, in that particular case it would also be helpful to clarify that you are specifically talking about AIME24). **I would need to see statistical tests included throughout the paper in order to consider increasing my score**.
* “Improved sampling efficiency alone cannot fully explain the observed gains” (line 267\) → Why not? I didn’t understand this argument, or how the subsequent experiments disprove it.
* “RL-tuned models show greater flexibility in following different plans compared to the base model” (line 285\) → how do you know they are actually following the plan rather than just ignoring it and doing their own thing?
* “Knowledge integration is a key driver of performance on harder tasks” (line 334\) → I’m not sure this conclusion is correct. It is hard to know for sure without seeing the distribution of difficulty, but based on the pass@1 rates it looks like there might be very few problems at higher difficulty levels (e.g. maybe just one at difficulty 10, and three at difficulty 9?). If so, the fact that the solve rate *looks* high might just be the result of high variance. It would be much better to (1) report the avg@8 score (as in the other plots), (2) to include error bars, and (3) to report how many questions there are at each difficulty level, in order to make sure the interpretations are correct.
* Some of the language used in the paper is overly broad given that only a single model (Qwen-2.5-Math-7B) was used. For example, the abstract says “RL-tuned models exhibit greater robustness… than their base counterparts” (line 11), but there is only one base model examined\! The language in the paper needs to be softened to clarify that Qwen 7B is the only model studied: I think this is ok because the benchmark is still interesting, but the claims in the paper need to be made more precise because without testing other models it is not possible to generalize the results in the paper to them.

### Missing details

* Significantly more detail is required to explain how SPARKLE is constructed. For example, which prompts are used to produce the plans/knowledge/decomposition? What prompts are used for the verification agent? How do you ensure that the subproblems all have correct answers and are solvable? How do you determine difficulty level? This doesn’t need to be in the main text, but at least should be in the appendix. **I would need to understand these details before considering raising my score.**
* I don’t understand how the “partial solution guidance” works. Could you please provide an example of what the prompt looks like? How do you choose the boundaries for the “chunks”? How is it different from methods for reverse curriculum training, e.g. [https://arxiv.org/pdf/2402.05808](https://arxiv.org/pdf/2402.05808) ? (Some of the details are scattered throughout the paper, e.g. where the reasoning traces come from—it would be really helpful to include a dedicated section in the appendix restating some of those things and explaining the whole procedure in more detail). **I would need to understand this better to consider raising my score.**
* What is the distribution of difficulty level (i.e. how many questions at each level)? It would be helpful to include a histogram in the appendix.
* Are the results reported in Figure 6 over the entire SPARKLE benchmark? It would be helpful to be more clear about this in the captions and text.

### Other questions/comments

* Please include error bars in Table 1 and all of the figures.
* In Figure 3, 4, & 5 (and in the appendix too) it would be helpful to include an “all” group which shows the average across all tasks.
* For the subproblem decomposition experiments, what is the average solve rate for the subproblems individually (not altogether)? And how many subproblems are there per problem (maybe you could include a histogram in the appendix)? It might be interesting to discuss your results in relation to [https://arxiv.org/pdf/2410.01748v1](https://arxiv.org/pdf/2410.01748v1) which I think has some similar findings.
* “RL tuning methodologies should prioritize high level reasoning strategies rather than detailed subproblem processing” (line 325\) → couldn’t you draw the opposite conclusion, i.e. that we should include more compositional training data to improve the models’ performance on solving compositional tasks like these?

**Ethical Concerns:**

["NO or VERY MINOR ethics concerns only"]

**Final Justification:**

My original concerns were: (1) lack of comparison to an SFT baseline, (2) no statistical tests, (3) missing details about how SPARKLE was created, and (4) missing details about partial solution guidance. In the rebuttal, the authors reported an analysis addressing (1), have promised to add (2), and have clarified (3) and (4). Thus all my primary concerns have been addressed and I recommend acceptance.

**Limitations:**

The paper should acknowledge that it only studies a single model (Qwen) and therefore it is unclear how well the results generalize.

**Quality:**

3

**Strengths And Weaknesses:**

Strengths:

* Clarity: I think the paper is structured well and it very clearly lays out the dimensions that are being studied.
* Significance: understanding RL reasoning performance at a more fine-grained level—beyond just accuracy on various benchmarks—is an important and burgeoning area of research.
* Originality: The various findings with Qwen 7B are very interesting and I believe will be of interest to the RLVF community: (1) RL performance can be boosted on hard problems if partial solutions are available; (2) performance of RL-trained models seems to be limited more by knowledge retrieval ability rather than planning ability; and (3) all models struggle with solving decomposed problems.

Weaknesses:

* Quality: there are a number of unsupported or overly broad claims. I detail these in the “Questions” section below, but will call out the two most important ones:
  * The paper is missing an important baseline, which is to compare to Qwen-2.5-7B-Math-Instruct. Some of the claims made in the paper about the base model may just be due to poor instruction-following capability, and it’s not clear whether the RL-trained models do better because they are better at instruction-following or because they’ve been trained with RLVF. Including an instruction-tuned baseline would help to clarify this.
  * There are no error bars or statistical tests reported in the paper, and some of the claims are made based on very small differences that could be due just to statistical error.
* Clarity: there are a number of missing details that make it difficult to fully evaluate the paper. I detail these in the “Questions” section below, but most importantly, there needs to be more detail about (1) how the SPARKLE dataset was constructed, and (2) how the “partial solution guidance” works.
* Quality: only a single model (Qwen-2.5-7B-Math) is studied in the paper, which limits the generality of the claims.

---

> ### Author Rebuttal · Authors · 2025-07-31
>
> We sincerely appreciate your thoughtful comments, detailed feedback, and support of our work! Due to word limit, we provide concise responses below. We are more than happy to elaborate further in the discussion phase.
>
> > Q1: Could the drop in base model performance stem from poor instruction-following? Please include Qwen-2.5-Math-7B-Instruct to isolate instruction tuning from reasoning.
>
> We did not include Qwen-2.5-Math-7B-Instruct in this work because, despite its name, it is **not a pure instruction-tuned model**. As noted in [1], it was trained by SFT and GRPO, incorporating additional CoT and tool-use supervision. This makes it difficult to disentangle instruction tuning from more general improvements in reasoning capability.
>
> Our focus is on comparing **base pretrained** vs. **RL-tuned** models to isolate the effect of RL on reasoning behavior. Notably, even the **base model shows substantial instruction-following ability**. In Fig. 2, scaling up the number of samples allows the base model to match RL-tuned model performance. On GSM8K, performance improves when planning information is provided—demonstrating plan-following capability even without SFT or RL (l.279–284, Fig.3).
>
> We will clarify this distinction in the final version and are happy to include results for Qwen-2.5-Math-7B-Instruct if the reviewer would find it helpful.
>
> > Q2&O1: Claims like “RL-tuned models achieve superior performance...” need statistical backing. Include error bars in Table 1 and figures.
>
> Thank you for this important suggestion. We have conducted detailed statistical analyses and updated the paper accordingly.
>
> To support the specific claim in l.289—that “RL-tuned models achieve superior performance when permitted to develop autonomous planning strategies”—we conducted a Welch’s t-test comparing SparkleRL Stage 2-aug with vs. without planning on AIME24. Results show: t = 3.38, p = 0.0052 (< 0.05), Cohen’s d = −1.69 (large effect size). This confirms the performance difference is statistically significant, supporting our conclusion.
>
> In addition, we now include mean ± standard deviation in Table 1 and have added error bars throughout the revised figures:
> |Model|AIME24|AMC23|MATH500|GSM8K|OlympiadBench|
> |-|-|-|-|-|-|
> |Qwen-2.5-Math-7B-SparkleRL-Stage 1|46.67% ± 2.11%|67.50% ± 1.58%|80.00% ± 0.13%|91.77% ± 0.12%|39.11% ± 0.21%|
> |Qwen-2.5-Math-7B-SparkleRL-Stage 2-aug|50.42% ± 0.99%| 71.25% ± 1.12%|81.00% ± 0.27%|92.38% ± 0.17%|40.11% ± 0.32%|
>
> We also ran Welch’s t-tests comparing Stage 1 vs. Stage 2-aug across all datasets:
> |Task|t|p-value|Cohen’s d|Sig.|
> |-|-|-|-|-|
> |AIME24|-3.81|0.0030|1.91|Yes|
> |AMC23|-4.58| 0.0006|2.29|Yes|
> |MATH500|-7.98|<0.0001|3.99|Yes|
> |GSM8K|-6.96|<0.0001| 3.48|Yes|
> |OlympiadBench|-6.15|<0.0001|3.07|Yes|
>
> These results confirm **performance improvements from Stage 1 to Stage 2-aug are statistically significant and robust across tasks**. We will incorporate these findings and the requested error bars in the final version.
>
> > Q3: Why not attribute the gains to improved sampling efficiency alone?
>
> We agree sampling efficiency can contribute to performance gains (as seen in Fig. 2). However, our goal is to explore *what additional capabilities RL may confer*. Key evidence:
> - Plan handling: RL-tuned models exhibit greater robustness and lower variance across planning variants (Sec.6.3, Fig.3).
> - Knowledge use: RL-tuned models can leverage external knowledge more effectively than base models (Sec.6.4).
>
> These differences suggest that RL affects **model behavior**, not just sampling variance. This motivates our SPARKLE analysis, which decomposes reasoning across plan following, knowledge use, and subproblem solving.
>
> > Q4: How do you know the model is following the plan rather than ignoring it?
>
> We define *plan-following flexibility* as the model’s ability to **interpret, adapt, or selectively use planning information**. All models receive identical prompts (see Fig.9), including: “Consider the following planning skeleton... you may adapt or extend this outline.”
>
> Under this setup, RL-tuned models are more robust and consistent (Fig. 3), implying they can integrate or disregard plans as appropriate. Base models show higher variance, often underperforming when given plans—suggesting lower flexibility or misalignment.
>
> > Q5: How are difficulty levels assigned? Are there enough high-difficulty samples to draw conclusions?
>
> We assign difficulty levels using AoPS Competition Ratings [2], which aligns with U.S. math competitions (e.g., AMC, AIME, USAMO, IMO). To avoid inductive bias, we *deliberately do not use pass@k to assign difficulty*--ensuring *objective and interpretable levels*.
>
> Since SPARKLE enriches existing benchmarks without creating any new problems, our dataset inherits their natural skew--only 4 problems are level 9–10. We avoid drawing conclusions from the tail and focus on levels 1–8, where sample sizes are more reliable (e.g., 40–41 examples at levels 7–8).
>
> At these levels:
> - *Knowledge augmentation* improves RL-tuned performance by *+4.9% (level 7)* and *+42.5% (level 8)*.
> - *planning* causes a *−4.9% drop (level 7)* and a *+15% gain (level 8)*.
> - Weighted across levels, knowledge yields an average gain of *+4.53%* vs. *+2.50%* from planning.
>
> These trends support our claim in l.334 that *knowledge integration is key to solving harder problems*. We will include difficulty histograms and clarify this in the revision.
>
> > Q6: Some claims are too broad given only one model was tested.
>
> We will revise the language to clarify this. We also include results for Qwen-2.5-32B in App. D.2, which exhibit consistent trends.
> We experimented with other open models (Mistral-7B, LLaMA-3.1-8B), but they fail to achieve non-trivial performance even after RL tuning (e.g., avg@32 0.2% on AIME24), making meaningful reasoning analysis infeasible. This further motivates our focus on models with strong mathematical priors.
>
> > Q7: What prompts are used for plan/knowledge/decomposition? How do you ensure correctness?
>
> SPARKLE is constructed using a pipeline (Sec. 3.2) where GPT-4.1 (with Internet access) is given: the original question, ground-truth answer, and expert step-by-step solution.
>
> We prompt it to generate:
> - Planning skeleton: 2–3 sentence outline of the reasoning path
> - Subproblem decomposition: step-by-step breakdown into well-posed, answerable subquestions
> - Knowledge extraction: key definitions, facts, or theorems (retrieved or inferred)
>
> Each annotation is then checked by a second GPT-4.1 verifier agent (prompted to assess correctness, completeness, coherence, pedagogical value). Human validators (math grad students) perform final verification.
>
> We will incorporate more of these implementation details in the revised appendix.
>
> > Q8: Please clarify how partial solution guidance works. How's it different from [3]?
>
> Our partial solution guidance augments selected training problems with intermediate solution chunks.
>
> Given a solution (excluding the final answer), we heuristically split it into 4 segments (e.g., by sentence) and create variants:
> - V1: question only
> - V2: question + chunk 1
> - V3: question + chunks 1–2
> - V4: question + chunks 1–3
> - V5: question + chunks 1-4
>
> This differs from [3] in two key ways:
> - We do **not enforce an easy-to-hard progression** or assume the model can solve subgoals in isolation.
> - We *train in a single stage (Stage 2)* with mixed supervision levels, rather than gradually increasing difficulty.
>
> Our method provides a clear and practical way for incorporating hard problems during RL. Even when the traces are human-written and misaligned with the model’s internal reasoning, RL-tuned models still learn useful strategies and show improved performance.
>
> > Q9&Q10: Is Fig 6 averaged over all tasks? Can you add an “All” group to Fig 3–5?
>
> Yes, Figure 6 reports results over the entire SPARKLE benchmark—we’ll clarify this in the caption and text. We will also add an “All Tasks” average to Figures 3–5 and include summary plots in the appendix.
>
> > Q11: What’s the subproblem-level solve rate? How does it relate to [4]?
>
> Thank you for this suggestion and we present the subproblem solve rate with average difficulty level for each task below:
> |Task|AIME24|AMC23|MATH500|GSM8K|OlympiadBench|
> |-|-|-|-|-|-|
> |Avg. Difficulty Level|3.6|2.1|1.5|1.0|3.1|
> |Subproblem Solve Rate|53.94%|72.96%|84.92%|95.60%|66.96%|
>
> We find that **subproblem solve rate drops as difficulty increases**, showing that decomposition alone is not sufficient to solve complex tasks—models still struggle with intermediate steps, especially under higher cognitive load. Most problems decompose into 4–6 subproblems; we’ll include histograms in the appendix.
>
> Compared to [4], our decompositions:
> - Are semantically aligned, derived from expert step-by-step solutions.
> - Span harder benchmarks beyond GSM8K (e.g., AIME, Olympiads).
> - Are studied under RL, which show that solving full problems does not imply solving subproblems (a gap unseen in prior work). This shows RL-tuned models may solve holistically, but are brittle under explicit decomposition.
>
> > Q13: Couldn’t your results suggest we need more compositional training data?
>
> Great question—we believe both directions are valid. However, our evidence suggests that *not all decompositions are helpful*. Rigid, human-written breakdowns can **hurt** model performance (e.g., −4.9% on Level 7), likely due to mismatch with internal solution paths.
>
> Thus, we argue for RL methods that emphasize **strategic flexibility**—the ability to adapt to guidance, retrieve knowledge, and reason through difficult tasks. Compositional training remains promising, but is suggested to align with model reasoning rather than impose rigid scaffolds. We’ll clarify this in the revision.
>
> [1] https://arxiv.org/abs/2409.12122
>
> [2] https://artofproblemsolving.com/wiki/index.php/AoPS_Wiki:Competition_ratings
>
> [3] https://arxiv.org/pdf/2402.05808
>
> [4] https://arxiv.org/pdf/2410.01748v1

---

> > ### Comment · Reviewer_1rRb · 2025-08-05
> > **Thanks for the rebuttal (1/2)**
> >
> > Thank you for the extensive rebuttal! I have a few further comments/questions.
> >
> > ### Q1: Instruction-following
> >
> > I think perhaps I didn't articulate my concern about instruction-following clearly enough. In the paper, you are concerned with "[isolating] the effect of RL on reasoning behavior" (quote from the rebuttal). However, reasoning isn't the *only* thing that can be improved via RL—it's also possible (likely) that RL improves instruction-following capability, especially if there is no prior SFT stage. Because the paper doesn't compare to any SFT models, it's not possible to know whether the gains from RL are due to improvements in **reasoning** or improvements in **instruction-following**. Without including an SFT/instruct baseline and analyzing the extent to which RL is improving reasoning vs. just instruction following, I don't think it's valid to claim anything about reasoning per se. Perhaps Qwen-2.5-Math-7B-Instruct is not exactly the right baseline for the reasons you mentioned, but I still think including some form of SFT/instruction-tuned baseline is important in distinguishing which of your results are due to better instruction following and which are due to better reasoning. (And just to be clear I read the response to reviewer wRv5 as well and your points about SFT there, which I think are fair—my suggestion here is not to show that RL is *better* than SFT but to use SFT as a way to identify what improvements are due to instruction following vs. reasoning. It's not the cleanest analysis in the world but essentially what I'm thinking is: SFT - baseline = improvements due to better instruction following, and RL - SFT = improvements due to better reasoning.)
> >
> > Alternatively, another way to address my concern would be to tone down the claims about reasoning in the text: e.g. if rather than saying "insights into how general-purpose (single-stage) RL affects reasoning across
> > a wide range of problem difficulties" (line 174), you replaced "reasoning" with "performance" (and similarly for many other places in the text where the word "reasoning" is used). And similarly, if you discussed in Section 6.3 that the robustness of the RL-trained models to incorporating the plan might be due to better instruction-following ability than the base model. I think it would be a lot more interesting to include an analysis trying to disentangle instruction-following and reasoning rather than changing the writing in the paper, but of course it's more work.
> >
> > (Also FWIW, I'm somewhat skeptical of drawing any strong conclusions based on GSM8K as it, and questions of similar style, have likely been included many times over in pretraining).
> >
> > ### Q2: Statistical tests
> >
> > Thanks for adding these! I guess because you're doing avg@8 the confidence are smaller than I was expecting—and it makes me much more confident about the results. I appreciate it.
> >
> > ### Q3: Sampling efficiency
> >
> > Thanks for the explanation, that's a nice insight. I think it would be helpful to readers to make this come through more strongly in the text (especially in section 6.2), as I clearly missed it.
> >
> > ### Q4: Plan following
> >
> > Thanks, that makes sense. Maybe a better way to phrase what you're looking at here is "plan integration" or "plan conditioning" rather than "plan following" per se, as "following" really implies to me something about the faithfulness of reasoning (as also brought up by reviewer wRv5).
> >
> > ### Q6: Other models
> >
> > Thanks for agreeing to revise the language, and for pointing to the 32B results (you should highlight these in the main text, I don't think they're mentioned at all!). These results are interesting especially in light of the instruction-following discussion above: all the 32B models seem to benefit from the plan information, which strengthens my belief that the robustness you see with the 8B RL-trained model is due to instruction-following improvements.

---

> ### Comment · Reviewer_1rRb · 2025-08-05
> **Thanks for the rebuttal (2/2)**
>
> ### Q11-Q13: Subproblems
>
> That's very interesting, thanks for providing that analysis! Are those numbers for SparkleRL-Stage 2?
>
> Re: citation [4], I mentioned it not as a suggestion to show how your results are different (it's clearly a different setup along multiple dimensions), but more that their findings could possibly help explain/augment your results too. They found that questions are *harder to solve in composition* rather than independently, which might also be an explanation for why asking the model to solve subproblems might make the performance worse in your case—even if the model *could* solve each subproblem independently, it might struggle to compose them. I suppose we can't fully identify the "independent" solve rate in your case but we can roughly compute $p$ as the average subproblem solve rate (what you've reported above) and then $p^n$ where $n$ is the number of subproblems, and compare that to the actual compositional solve rate (what was already in the paper). So based on the numbers you reported above:
>
> |                                   | AIME24    | AMC23     | MATH500   | GSM8K     | OlympiadBench |
> | --------------------------------- | --------- | --------- | --------- | --------- | ------------- |
> | Expected compositional solve rate | 10.8%     | **51.5%** | **78.2%** | **95.6%** | 28.8%         |
> | Actual compositional solve rate   | **17.5%** | 37.5%     | 65.8%     | 87.6%     | **34.4%**     |
>
> So from this rough analysis we can see that on AMC23, MATH500, and GSM8K you get the same results as [4]—the compositional solve rate is lower than what you'd expect than if you were solving each problem independently, perhaps suggesting something about the difficulty of solving the subproblems *in composition*. But interestingly on AIME24 and OlympiadBench the results go the other way! These are the hardest problems, and have probably been overfit to less in pretraining, so perhaps in this case the decomposition could help, *if* the model were better at solving the individual subproblems (which it seems quite poor at).
>
> ### Q5, Q7-Q10
>
> Thanks for the clarifications, these address my concerns.

---

> > ### Author Response · Authors · 2025-08-07
> >
> > Thank you very much for your thoughtful and detailed feedback. We are glad that Q2–Q10 addressed your concerns. We agree with the subtleties you described around reasoning vs. instruction-following and we will qualify/tone-down the language in the text of the draft around this. We provide additional information for Q1 and Q11–Q13 below.
> >
> > > Q1: instruction following
> >
> > Great point! We ran the suggested SFT experiment. We performed SFT on the base model (Qwen-2.5-Math-7B) using CoT traces on hard questions, followed by RL finetuning. The results are shown below:
> >
> > | Model | AIME24 | AMC23 | MATH500 | GSM8K | Olympiad Bench | Avg. |
> > | -------------------------------- | ---------- | ---------- | --------- | ------ | -------------- | ------ |
> > | Qwen-2.5-Math-7B | 16.67% | 42.50% | 44.03% | 44.30% | 28.65% | 35.23% |
> > | Qwen-2.5-Math-7B (add.plan) | 9.58% | 30.94% | 41.45% | 46.92% | 18.85% | 29.55% |
> > | **Qwen-2.5-Math-7B-SFT**| 15.42% | 50.63% | 68.55% | 85.03% | 31.44%        | 50.21% |
> > | **Qwen-2.5-Math-7B-SFT (add. plan)** | 14.17% | 50.31% | 69.08% | 88.84% | 31.78%        | 50.83% |
> > | **Qwen-2.5-Math-7B-SFT+RL**  | 33.33% | 64.06% | 77.33% | 91.58% | 36.74%        | 60.61% |
> > | **Qwen-2.5-Math-7B-SFT+RL (add. plan)** | 36.25% | 67.19% | 79.45% | 94.47% | 39.80%        | 63.43% |
> >
> >
> > What we observe is that
> >
> > - Overall, as expected, SFT performs better than base, but worse than RL
> > - In terms of plan-following flexibility, **SFT is not necessarily able to use a provided plan**---just like the base model, and unlike the RL-tuned model
> >
> > This finding suggests that RL improvements (at least in terms of plan use flexibility) **cannot be solely attributed to improved instruction following**.
> >
> > We agree that this is a very interesting direction! We will add the discussion above to our draft.
> >
> > > Q11-13: Subproblems
> >
> > Yes, these numbers are reported for SparkleRL-Stage 2. Very interesting findings, and we appreciate your detailed explanation. Thank you for sharing these observations—they are helpful for further understanding the role of problem decomposition and problem difficulty in mathematical reasoning under RL.

---

> > > ### Comment · Reviewer_1rRb · 2025-08-07
> > > **Response**
> > >
> > > Thank you for running these experiments with SFT! These are very interesting, and I agree it shows that RL is bringing something to the table above and beyond simple instruction following. (I am rather surprised though that SFT+RL is so much worse than just RL, though... it is probably worth trying to understand why this is for the final version of the paper.)
> > >
> > > I think with this all of my concerns are addressed. I am happy to increase my score.

---

### Official Review · Reviewer_wRv5 · 2025-07-03

**Clarity:** 2
**Significance:** 2
**Originality:** 2
**Rating:** 3
**Confidence:** 4

**Summary:**

The paper presents a framework for analysing the reasoning behaviours of LLMs along three dimensions: planning and execution, knowledge utilisation, and chain of subproblems. The authors also explore the use of curricula and solution guidance to improve the learning potential of hard problems when conducting RL for reasoning. Results demonstrate that these approaches improve the utility of hard problems during training and reveal that RL training reliably enhances two out of the three reasoning behaviours considered.

**Questions:**

Why does performance by difficulty level have a U-shape with knowledge augmentation?

**Ethical Concerns:**

["NO or VERY MINOR ethics concerns only"]

**Final Justification:**

The paper makes progress towards quantitatively evaluating aspects of reasoning models besides their final performance, but the categorisation of three particular types of behaviours seems limited and is not likely to be of significant impact in its current state.

**Limitations:**

Limitations are discussed in an Appendix, but should feature in the main text.

**Paper Formatting Concerns:**

No formatting concerns.

**Quality:**

2

**Strengths And Weaknesses:**

Strengths:

By not focusing on benchmark accuracy alone, the paper reveals some qualitative behaviours, which can be quantitatively measured, that are enhanced by RL finetuning. These insights provide intuitive insights into the general capabilities that translate to improved performance on reasoning tasks. The results also demonstrate that the proposed problem difficulty curriculum and solution augmentation do indeed improve learning from hard problems.

Weaknesses:

The dual focus of analysing reasoning behaviours and improving learning from hard problems is disjointed. The proposed framework is also not comprehensive, as it fails to account for aspects such as the faithfulness of reasoning. While the paper reveals that problem decomposition is not useful, this unintuitive result warrants further investigation, or could even motivate an approach to making better use of problem decomposition. The impact of additional training is not accounted for; comparisons to SFT would and other offline methods such as DPO would be necessary to robustly show that it is RL training specifically that reinforces these behaviours. Though not necessarily a weakness, the findings of the paper are not particularly surprising and so their impact may be minimal.

---

> ### Author Rebuttal · Authors · 2025-07-31
>
> We appreciate your comments and suggestions and we address each question and provide further clarifications in detail below. Please let us know if you have any additional questions or comments and we are more than happy to discuss further.
>
> > W1: The dual focus of analyzing reasoning behaviors and improving learning from hard problems is disjointed.
>
> The dual aspect here **is intentional; in fact, we believe it is crucial for our overall findings**. Our work is centered on analyzing mathematical reasoning under reinforcement learning (RL). When applying RL to a set of problems, the key knobs are (i) the components necessary to solve a problem (planning, knowledge, etc.,) and (ii) the raw difficulty of the problems. A full characterization of RL for reasoning is not possible without addressing both of these elements.
>
> Our Hard-Augmented Stage 2 setting is not proposed as a general solution to the reasoning challenges observed, but rather as a complementary contribution: it demonstrates a practical method for leveraging hard problems in RL training—an area prior work has found difficult due to absent reward signals. Together, our dual contributions—(1) analysis framework and benchmark of reasoning behaviors across planning and execution, knowledge utilization, and chain of subproblems; and (2) a method to incorporate hard examples—provide stronger empirical evidence for understanding the effect of RL. The availability of two distinct RL-tuned models (SparkleRL-Stage 1 and SparkleRL-Stage 2-aug) allows us to conduct more fine-grained analysis than is typically possible in existing work.
>
> > W2: The proposed framework is also not comprehensive, as it fails to account for aspects such as the faithfulness of reasoning.
>
> We appreciate this observation. Our work focuses on three core aspects of mathematical reasoning captured by the SPARKLE framework: plan-following, knowledge utilization, and subproblem decomposition. While faithfulness is indeed an important dimension, it is orthogonal to the goals of our current study. We believe that a fine-grained and dedicated analysis of these three components is equally valuable, as they represent foundational building blocks of mathematical reasoning.
>
> Our goal is not to exhaustively cover all aspects of reasoning—necessary trade-offs must be made to support a *focused investigation* with a clear central theme. We hope our work can serve as a cornerstone for future research to extend the framework, including incorporating dimensions such as faithfulness.
>
>
> > W3: While the paper reveals that problem decomposition is not useful, this unintuitive result warrants further investigation, or could even motivate an approach to making better use of problem decomposition.
>
> We would like to clarify that our paper **does not claim that problem decomposition is not useful**. Instead, our findings reveal a **more nuanced perspective** on how decomposition interacts with RL-tuned models.
>
> In our analysis of plan-following (Sec. 6.3), we observe that **RL-tuned models exhibit greater flexibility in generating and executing their own plans**. Interestingly, even valid human-written plans do not always yield improved performance—particularly on harder benchmarks like AIME24—because high-level plans may overlook critical edge cases. However, for simpler datasets such as GSM8K (difficulty level 1 out of 10), explicit plans consistently improve performance for both base and RL-tuned models. In these cases, planning helps reduce cognitive load by structuring the reasoning into manageable steps.
>
> In our subproblem decomposition analysis (Sec. 6.5), we consider a problem unsolved if any subproblem is answered incorrectly. Under this strict criterion, RL-tuned models often struggle, especially when the decomposition is overly granular. Combined with our plan-following results, this suggests that **fine-grained decompositions may disrupt the model’s internal reasoning flow**, rather than supporting it.
>
> We view this as an valuable direction for future research: rather than relying on human-authored decompositions, it could more effective to design model-aligned decomposition strategies that are better matched to the model’s inductive biases. However, proposing a new algorithm that leverages this phenomenon would require dedicated study and is beyond the current scope of our work.
>
> > W4: The impact of additional training is not accounted for; comparisons to SFT and other offline methods such as DPO would be necessary to robustly show that it is RL training specifically that reinforces these behaviors.
>
>
> Thanks for bringing this up! We would like to clarify that our paper does not evaluate or include SFT or DPO methods **by design**. To avoid introducing confounding effects, we deliberately focus on comparing base pretrained models and RL-tuned models, isolating the impact of RL. This design choice complements prior studies that specifically analyze the interplay between SFT and RL [1,2] (see l.194–195).
>
> Therefore, it is **not** our objective to claim that RL, rather than SFT or DPO, **uniquely** reinforces these behaviors. Instead, our goal is to provide a focused, fine-grained analysis of reasoning behaviors that emerge specifically from RL training.
>
>
> To ensure a clean comparison, we initialize both the base and RL models from the same pretrained checkpoint (Qwen-2.5-Math-7B) and apply RL training only, without any intermediate supervised fine-tuning (SFT) or preference optimization (e.g., DPO). This setup allows us to **attribute the behavioral differences directly to RL**, without contamination from other training paradigms.
>
> While comparisons to SFT or DPO are indeed important in the broader landscape, they are **outside the scope of this work**. Our goal is not to establish whether RL is superior to SFT or DPO, but rather to provide a **fine-grained analysis of how RL shapes reasoning behaviors**—an area that remains underexplored despite RL's widespread use in modern LLM pipelines.
>
> We hope this clarifies our motivation and experimental design. We believe that narrowing the focus to RL is not a limitation but a necessary step toward understanding its unique contributions.
>
>
> [1] Chu et al., Sft memorizes, rl generalizes: A comparative study of foundation model post-training, 2025
> [2] Yeo et al., Demystifying long chain-of-thought reasoning in llms, 2025.
>
>
> > Q: Why does performance by difficulty level have a U-shape with knowledge augmentation?
>
> Thank you for this thoughtful question. The observed U-shaped trend in Fig. 11 primarily reflects the **natural imbalance in difficulty levels within existing math benchmarks**. Our framework is designed to **augment** existing problems—*not to construct new ones or rebalance the dataset*—so it inherits a skewed distribution, with only 4 total problems at levels 9–10. This sparsity at the top makes accuracy volatile and can visually exaggerate trends like the U-shape. We will clarify this in the revised manuscript.
>
> However, **this pattern does not contradict our central finding**: *knowledge augmentation is more beneficial than planning, especially for harder problems*. For example:
> - On level 8 (40 problems), knowledge augmentation improves performance by **+42.5%**, compared to a **+15%** gain from planning.
> - On level 7 (41 problems), knowledge gives a **+4.9%** gain, while planning causes a **−4.9%** drop.
> - When weighted averaged across all levels, knowledge yields a **+4.53%** gain, compared to **+2.50%** from planning.
> These trends remain robust outside the sparsely populated level 9–10 range.
>
> These trends are *robust across sufficiently sampled levels (1–8)* and support our conclusion that *external knowledge access is a key driver of performance on complex tasks*(l.334).
>
> We also emphasize that difficulty levels are assigned based on AoPS Competition Ratings [[AoPS](https://artofproblemsolving.com/wiki/index.php/AoPS_Wiki:Competition_ratings?srsltid=AfmBOopoJxcbOKLhC-4La5S32ov_LvXuDDy5BLb-45hGUO9Q0pxDTh2Z)], which map to the structure of major U.S. math competitions: Levels 1–2 correspond to AMC 8, early AMC 10 and AMC 12 problems (e.g., standard middle school math), Levels 4–6 span through later AMC 12, AIME and intro level Olympiad-styple problems, and Levels 7–10 include USAMO, IMO, and other hard Olympiad-level problems.
>
> We *deliberately avoid* using metrics like pass@k of model performance to define difficulty levels, as they may introduce model-induced inductive bias. Instead, we rely on competition-based standards to provide an objective and interpretable standard. While pass@k filtering would allow easier balancing, it introduces model-induced inductive bias, which we explicitly avoid.

---

> > ### Comment · Reviewer_wRv5 · 2025-08-05
> >
> > Thank you for your rebuttal. As it addresses several of my concerns, I have increased my score. However, given that this paper's core contribution is an analysis of reasoning models, I still find the three chosen factors to be a somewhat crude categorisation of the possible behavioural variances.

---

> > > ### Author Response · Authors · 2025-08-06
> > >
> > > Thank you for your updated rating and follow-up comment!
> > > We would like to take this opportunity to further clarify the rationale behind our design choices and the scope of our work:
> > >
> > > **1. On the rationale behind the three dimensions**
> > >
> > >  Our goal is not to comprehensively categorize all possible LLM reasoning behaviors—a task far beyond the scope of a single paper—but to focus on a core set of foundational dimensions that are widely recognized as important to mathematical reasoning. Specifically, we center our analysis on knowledge utilization [1,2,3], subproblem decomposition [4,5,6,7], and planning [8]. While we acknowledge that this selection is non-exhaustive, we chose to prioritize these crucial dimensions.
> > >
> > > Producing a fine-grained categorization of reasoning and the factors behind it is an interesting idea--and something we have been thinking about as well! We are inspired by research in cognitive science [9–12], which identifies knowledge retrieval, planning, and decomposition as some of the most important components of human problem-solving. We will add this motivation for extra clarity in our updated draft.
> > >
> > > **2. Contributions beyond analysis framework**
> > >
> > > In addition to the analysis framework, we would also like to highlight two further contributions:
> > > - SPARKLE benchmark, a fine-grained benchmark with 2,564 open-ended questions spanning 10 difficulty levels and 9 domains, with rigorous human annotation and validation pipeline.
> > > - Partial-solution scaffolding, a new strategy for multi-stage curriculum RL that provides an effective way to utilize the hardest problems—often accompanied by noisy or mixed-quality solution traces—in second-stage RL training, leading to further improvements in LLM reasoning.
> > >
> > > We appreciate your engagement and feedback! We are happy to go into more detail or answer any additional questions you may have.
> > >
> > >
> > > References:
> > >
> > >  [1] Ko et al., Hierarchical Deconstruction of LLM Reasoning: A Graph-Based Framework for Analyzing Knowledge Utilization, EMNLP 2024
> > >
> > >  [2] Wang et al., Boosting Language Models Reasoning with Chain-of-Knowledge Prompting, ACL 2024
> > >
> > >  [3] Gao et al., Retrieval-Augmented Generation for Large Language Models: A Survey, 2024
> > >
> > >  [4] Tomer et al., Break it down: A question understanding benchmark, ACL 2020
> > >
> > >  [5] Zhou et al., Least-to-most prompting enables complex reasoning in large language models, ICLR 2022
> > >
> > >  [6] Dua et al., Successive prompting for decomposing complex questions, EMNLP 2022
> > >
> > >  [7] Khot et al., Decomposed prompting: A modular approach for solving complex tasks, ICLR 2023
> > >
> > >  [8] Wei et al., PlanGenLLMs: A Modern Survey of LLM Planning Capabilities, 2025
> > >
> > >  [9] Allen & Herbert, Human Problem Solving, 1972
> > >
> > >  [10] Peter & Philip, Psychology of Reasoning: Structure and Content, 1972
> > >
> > >  [11] Ken Manktelow, Reasoning and Thinking, 2012
> > >
> > >  [12] Jiří Dostál, Theory of Problem Solving, 2015

---

### Official Review · Reviewer_SZLe · 2025-07-09

**Clarity:** 4
**Significance:** 3
**Originality:** 3
**Rating:** 5
**Confidence:** 4

**Summary:**

This paper proposes a number of methods for analyzing strengths and weaknesses of language model reasoning, and uses these methods to explore the effect of RL training (through GRPO) on reasoning in math domains. They also propose and evaluate a method for training through RL on difficult problems based on providing partial solution scaffolds.
- The first angle they explore deals with planning versus executing – they test models with and without plans being provided, and surprisingly find that providing plans can often actually hurt the performance of models, especially base (non-RL) models, presumably because models are better at following the plans that they come up with instead of provided plans.
- The second angle they study is on the ability to recall relevant background knowledge – they test models with and without relevant theorems/definitions/etc being provided in the prompt, and find RL based methods are much more strongly able to make use of provided knowledge (in fact it often hurts the base models!).
- The third angle looks into the problems with providing plans (the first angle) in more detail by exploring how well the model can solve a subproblem given the ground truth solution prefix up to that subproblem. They find that models are far better at solving overall problems than solving the subproblem sequence.

**Questions:**

- Relating to both 6.3 and 6.5, I'm curious if you have any qualitative observations from looking at what sorts of human-provided plans (6.3) or subproblems (6.5) are difficult for the model to work with. Do you have any observations on how these plans and subproblems differ from those that the model comes up with and prefers to work with?
- On my minor weakness about the Hard-Augmented results, it'd be helpful to hear a bit more of what kind of settings motivate this, or where you see this result being useful?

**Ethical Concerns:**

["NO or VERY MINOR ethics concerns only"]

**Final Justification:**

Maintaining my Accept rating – my minor concerns were resolved in the rebuttal.

**Limitations:**

Yes.

**Quality:**

4

**Strengths And Weaknesses:**

Overall I'm very positive on this paper, I think that it's an excellent fit for NeurIPS. It is beautifully written and presents a series of interesting results pertaining to a very hot area in the field right now (effects of RL on language models). The paper reads like an excellent example of good science – clear questions, well-designed experiments, beautifully presented. I'm not an expert in RL for LLMs and may not know all the related work, but given my understanding this is a solid, original contribution to the field.

Strengths
- I find the overall framework compelling. It's true that we need to move beyond accuracy based metrics, and I think that this is a nice breakdown into different aspects of the problem (planning, background knowledge, ability to execute plans).
- The dataset construction pipeline including a final check by graduate students with a math background is an excellent step of verification and one that many papers don't do. And likewise during the hard-solution curation there's a graduate student verification step to find flawed solutions.
- Section 6 is a nice presentation of a number of interesting results. The ordering and flow of the section made it quite easy to read.
    - The results in 6.1 on training on scaffolded problems shows that the proposed method provides a reasonable boost. However I feel the main contribution of the paper is primarily in the other results.
    - The result in 6.3 that externally provided plans decrease performance for many base/RL models over many domains (but that RL models are more robust to this) is particularly interesting – the idea that actually models are particularly good at following their own plans, and being forced into an externally provided plan can decrease performance. I appreciated the granular explanation of why that trend doesn't hold for GSM8K.
    - The section 6.4 result that external knowledge harms the base model but helps the RL models was also really interesting, as was the 6.5 results on models having more trouble solving all subproblems than overall tasks.
    - 6.6 was an important analysis to include for shedding a bit more light on the 6.3/6.4 result trends, showing how they vary with problem difficulty.
- The writing of Section 3.1 is excellent, each of the axes is presented quite clearly. The style works well – leading with a sentence about the more general reasoning phenomenon the axis is trying to capture then getting very concrete and specific. And the high quality writing carries through the rest of the paper. The figures are also beautiful, the paper overall has a very high level of polish.

Weaknesses
- I don't have any strong weaknesses.
- The Hard-Augmented model results (6.1) are interesting but don't feel too significant. In part because the approach requires having solutions (or partial solutions) to these problems, at which point one could just train on those solutions in the first place.

Minor edits and suggestions
- line 208: difficulty -> difficult
- Figure 6 isn't labelled as "Figure 6" anywhere
- Figure 6a caption: "stable" -> "stability"
- I wonder if 6.6 could be moved up to be right after 6.4, since it's about how 6.3/6.4 vary with difficulty? Possibly 6.5 could fit well right after 6.3, since it's a bit of a look in detail into how things break down when solving human-provided problems. Just a thought / suggestion though, may or may not be better.

---

> ### Author Rebuttal · Authors · 2025-07-31
>
> Thank you for your valuable feedback, thoughtful comments, and your support of our work! We address each question in detail below. Please let us know if you have any additional questions or comments and we are more than happy to discuss further.
>
> > Q1: Any qualitative observations from looking at what sorts of human-provided plans (6.3) or subproblems (6.5) are difficult for the model to work with. Do you have any observations on how these plans and subproblems differ from those that the model comes up with and prefers to work with?
>
> Yes, we obtained several qualitative observations while comparing model-generated reasoning with human-provided plans and decompositions.
>
>
> **Planning:**
>
> A common pattern is that **high-level human plans often lack sufficient granularity, particularly for more difficult problems**. This can lead models to overlook important edge cases, especially in mathematically nuanced settings.
>
> For example, in Table 3 of the supplementary materials (AIME24), the provided planning instruction states: "*Identify the pattern that arises for losing positions.*" Following this plan, the model correctly identifies only one of the two correct losing scenarios (n ≡ 2 (mod 5)), but misses the other (n ≡ 0 (mod 5)), leading to an incorrect final answer.
>
> In contrast, when no plan is provided, the same model generates Python code to perform dynamic programming and simulates the game for all n ≤ 2024. This strategy leads to the correct result (809), successfully capturing both losing scenarios.
>
> This example illustrates a broader trend: RL-tuned models often perform better when allowed to explore the problem space using their internal reasoning mechanisms (e.g., generating code or applying strategies they are familiar with), rather than following externally imposed plans. While human-written plans can be helpful for simpler tasks (e.g., GSM8K), they may restrict model flexibility or bias the model toward **unfamiliar** or **suboptimal** strategies on more challenging tasks.
>
> **Subproblem decomposition:**
>
> We observe a similar pattern when studying subproblem decomposition (Sec. 6.5). While human-provided subproblems are often semantically meaningful, breaking a problem into isolated steps can disrupt the model’s ability to reason holistically. In the same AIME24 example, a corresponding subproblem is phrased as:
>
> Q: What is the pattern of losing positions for n tokens?
>
> Despite being logically sound, the model still fails to identify one of the two key scenarios. This suggests that isolated subproblems can inherit the same brittleness as high-level plans if they are not grounded in the model’s natural reasoning strategies.
>
>
> > Q2: What motivates the Hard-Augmented setting and the possiblity of training on these solutions.
>
> Thank you for the thoughtful question! We are motivated by prior work that suggests RL methods with outcome rewards such as GRPO *cannot benefit from the hardest problems due to the lack of positive reward signals* (i.e., all trajectories in a group yield wrong answers). We explore whether these hard problems could still be leveraged *in a way that improves performance*—without compromising the capabilities learned by a well-tuned RL model from Stage 1.
>
> We agree that a natural approach would be to train on available solution traces with supervised fine-tuning (SFT). However, SFT does not yield promising results in our setting for the following reasons:
>
> - The training set used in Stage 1 contains over 40K math problems, where many solution traces are noisy (e.g., incomplete solutions). These traces vary in quality---from simplistic chain-of-thought outputs to human-written final answers. While obtaining new traces by distilling from competitive LLMs is possible, it would require substantial compute and additional human validation to ensure correctness and clarity.
> - Empirically, we conducted an additional experiment by applying SFT on these noisy traces. The results are shown below:
>
> | Model                     | AIME24     | AMC23      | MATH500    | GSM8K      | OlympiadBench |
> | ------------------------- | ---------- | ---------- | ---------- | ---------- | ------------- |
> |**SparkleRL-Stage 1**| 46.67%| 67.50%| 80.00%     |91.77%|39.11%|
> |**SparkleRL-Stage 2-aug** | **50.42%**| **71.25%**| **81.00%**|**92.38%**| **40.11%**|
> |**SFT on hard problems with solutions**|15.00%|53.44%|70.03%|88.30%|30.70%|
>
> We observe that SFT on hard problems from the SparkleRL-Stage 1 model leads to **significant performance degradation** across benchmarks. In contrast, our **Stage 2-aug** method yields consistent improvement.
>
> The key distinction in Stage 2-aug (see L202–216) is that we augment the hardest unsolved examples from Stage 1 with partial solutions provided in the input prompt, rather than full solutions used for fine-tuning. This design preserves on-policy learning, which is important for effective RL training, thereby further boosting the reasoning performance.
>
> > Typos
>
> Thank you! We will carefully revise and correct all typos in the final revision.

---

> > ### Comment · Reviewer_SZLe · 2025-08-05
> >
> > Thank you for the rebuttal, that resolves my questions and my minor concern.

---

### Note · Authors · 2025-08-13

We sincerely thank all reviewers for their time and effort in providing valuable feedback and suggestions. We are encouraged by R1’s recognition of our work as “an excellent example of good science–clear questions, well-designed experiments, beautifully presented", that it "presents a series of interesting results" and as “a solid, original contribution to the field.” We are also glad that reviewers acknowledged the importance of our fine-grained approach to understanding RL’s impact on reasoning beyond accuracy metrics (R1,R3) and found the results interesting and relevant to the RLVF community (R3). We value R2’s note that our work offers insights into general capabilities that boost reasoning performance, with results showing that our curriculum and augmentation improve learning from hard problems. We are pleased that reviewers found our presentation high-quality, well-structured, and easy to read (R1,R3,R4).

We have addressed reviewers’ feedback. We'd like to highlight:
- **Motivation for the three evaluation dimensions**: We explicitly connect our choice of planning, knowledge utilization, and subproblem decomposition to cognitive science literature identifying these as core components of human problem-solving.
- **Contributions beyond the analysis framework**: We clarified that our work also introduces:
  - **SPARKLE benchmark**: 2,564 rigorously human-verified, open-ended math problems spanning 10 difficulty levels and 9 domains, annotated with plans, relevant knowledge, and subproblems.
  - **Partial-solution scaffolding**: a new multi-stage curriculum RL strategy that effectively leverages the hardest problems—even those with noisy or mixed-quality traces—in second-stage RL training, yielding further improved performance.
- **Human validation**: Our benchmark is constructed under a multi-stage pipeline with final human validation (Sec 3.2).
- **Statistical rigor**: We added error bars, standard deviations, and Welch’s t-tests across all main results, confirming that our observed gains are statistically significant and robust.
- **Study on SFT**: While our focus is on RL, we conducted additional experiments with (1) Stage 2 on SFT, along with rationale for why SFT is not used in Stage 2; and (2) SFT and SFT+RL from the base model. These results confirm that RL brings the benefits characterized in the paper, with behaviors that are qualitatively different from those produced by SFT.

We have updated our manuscript to incorporate the discussions.

---

### Decision · Program_Chairs · 2025-09-17

**Decision:**

Accept (poster)

**Comment:**

This paper investigates the benefits of RL training. In particular, the paper explores these benefits in the context of plan-following and execution, problem decomposition, and (3) improved reasoning and knowledge utilization. Experimental evidence is provided for each of these aspects. Interestingly, the paper shows that plans derived from human solutions are not necessarily good for the model to execute, which is not conceptually surprising. The paper also considers the effect of training exclusively on hard problems and using hard problems augmented with partial solutions and provides interesting insights. The reviews for the paper are mixed. While the problem setting is interesting, the reviewers were concerned about insufficient ablations, novelty of the findings and overall presentation of the paper.

After going through the paper, it is somewhat surprising that the main body almost omits the key details about the experiments (e.g. details about the plans and how they are used). There are many design choices involved in the experiments and careful review of them is required to ensure robustness of these findings across settings (especially with respect to different design choices). I'm happy that this discussion came during the rebuttal process. It is important to address these concerns before publication of the final version because in the current form the paper seems somewhat misleading. It is important to clearly highlight the limitations of the work raised during discussion with Reviewer wRv5 and Reviewer 1rRb.